# Gene regulatory mechanisms guiding bifurcation of inhibitory and excitatory neuron lineages in the mouse anterior brainstem

Sami Kilpinen*, Lassi Virtanen, Silvana Bodington Celma, Amos Bonsdorff, Heidi Heliölä, Kaia Achim*, Juha Partanen*

Molecular and Integrative Biosciences Research Programme, Faculty of Biological and Environmental Sciences, University of Helsinki, Helsinki, Finland

## eLife Assessment

This work is a **important** resource for hypothesis testing of candidate upstream transcriptional regulatory factors that control the spatiotemporal expression of selector genes and their targets for GABAergic vs glutamatergic neuron fate in the anterior brainstem. Extensive high-quality datasets were generated and state of the art computational methods were **convincingly** implemented to identify candidate regulatory elements. The work will be of interest to biologists working to understand neuronal gene regulatory networks.

**\*For correspondence:**
sami.kilpinen@helsinki.fi (SK);
kaia.achim@helsinki.fi (KA);
juha.m.partanen@helsinki.fi (JP)

**Competing interest:** The authors declare that no competing interests exist.

**Abstract** Selector transcription factors (TFs) control choices of alternative cellular fates during development. The ventral rhombomere 1 of the embryonic mouse (*Mus musculus*) brainstem produces neuronal precursors that can differentiate into either inhibitory GABAergic or excitatory glutamatergic neurons important for the control of behaviour. TFs *Tal1*, *Gata2*, and *Gata3* are required for adopting the GABAergic neuronal identity and inhibiting the glutamatergic identity. Here, we asked how these selector TFs are activated and how they control the identity of the developing brainstem neurons. We addressed these questions by analysing chromatin accessibility at putative gene regulatory elements active during GABAergic and glutamatergic neuron lineage bifurcation, combined with studies of TF expression and DNA binding. Our results show that the *Tal1*, *Gata2*, and *Gata3* genes are activated by highly similar mechanisms, with connections to regional patterning, neurogenic cell cycle exit and general course of neuronal differentiation. After activation, *Tal1*, *Gata2*, and *Gata3* are linked by auto- and cross-regulation as well as regulatory interactions with TFs of the glutamatergic branch. Predicted targets of these selector TFs include genes expressed in GABAergic neurons, glutamatergic neurons, or both. Unlike genes specific to the glutamatergic branch, the genes expressed in GABAergic neurons appear to be under combinatorial control of *Tal1*, *Gata2*, and *Gata3*. Understanding gene regulatory interactions affecting the anterior brainstem GABAergic and glutamatergic neuron differentiation may give genetic and mechanistic insights into neurodevelopmental traits and disorders.

## Introduction

The anterior brainstem is a complex neuronal mosaic controlling mood, motivation and movement. These neurons include diverse inhibitory GABAergic and excitatory glutamatergic neurons, some of which are associated with the dopaminergic and serotonergic systems.

An important source of these neurons is the ventrolateral rhombomere 1 (r1), where GABA- and glutamatergic neuron precursors are generated from a neuroepithelial region expressing a homeodomain (HD) transcription factor (TF) *Nkx6-1* (*Lahti et al., 2016*; *Waite et al., 2012*). We here refer to these precursors as rhombencephalic V2 domain (rV2), reflecting its molecular similarity to the V2 domain in the developing spinal cord. The rV2 precursors give rise to GABAergic neurons in the posterior Substantia Nigra pars reticulata, Ventral Tegmental Area, Rostromedial Tegmental Nucleus, and Ventral Tegmental Nucleus, all important for various aspects of behavioural control (*Lahti et al., 2016*; *Morello et al., 2020a*). A failure in the differentiation of the rV2 GABAergic neurons in the embryonic mouse brain, caused by mutation of the *Tal1* gene, results in robust behavioural changes in postnatal mice, including hyperactivity, impulsivity, hypersensitivity to reward and learning deficits (*Morello et al., 2020a*).

The *Nkx6-1* positive proliferative progenitors of rV2 give rise to postmitotic V2b-like GABAergic precursors expressing selector genes *Tal1*, *Gata2*, and *Gata3*, and V2a-like glutamatergic precursors expressing *Vsx1* and *Vsx2* (*Achim et al., 2014*; *Joshi et al., 2009*; *Muroyama et al., 2005*). In the absence of *Tal1*, *Gata2*, and *Gata3* function, differentiation of all the rV2 precursors is directed to glutamatergic neuron types (*Achim et al., 2012*; *Lahti et al., 2016*). *Tal1*, *Gata2*, and *Gata3* also retain their expression in the mature GABAergic neurons. Thus, Tal1, Gata2, and Gata3 TFs operate as fate selectors, possibly responsible for both the induction and maintenance of the GABAergic identity in the anterior ventrolateral brainstem. It has remained unclear how the selector TFs are activated during development and how they guide neuronal differentiation in the anterior brainstem. To address these questions, we have characterized the accessibility and TF binding in putative gene regulatory elements and correlated these events with changes in gene expression during differentiation of GABAergic and glutamatergic neurons in the anterior brainstem. Our results suggest that the GABAergic selector TF genes are activated by interconnected mechanisms reflecting developmental regionalization and timing, and that their functions converge to promote GABAergic and suppress glutamatergic neuron differentiation.

## Results

### Transcriptomic dynamics and chromatin accessibility in differentiating rV2 GABAergic and glutamatergic neurons in mouse embryo

To understand the sequence of gene expression changes that correlate with the bifurcation of the rV2 GABAergic and glutamatergic cell lineages, we first studied the transcriptome and the genome accessibility in developing rV2 precursors. We used data from the scRNA-seq of the ventral r1 region of E12.5 embryos (*Morello et al., 2020a*), where the rV2 lineages were identified based on the expression of markers of common proliferative progenitors (*Ccnb1*, *Nkx6-1*) and markers of the post-mitotic GABAergic (*Tal1*, *Gad1*) and glutamatergic precursors (*Nkx6-1*, *Vsx2*, *Slc17a6*) (*Figure 1A, C*, *Figure 1—figure supplement 1A*, *Supplementary file 1*). We found that the rV2 cells were arranged on pseudotemporal axes representing differentiating GABAergic or glutamatergic lineages (*Figure 1A, B*). The GABAergic and glutamatergic selector TF genes (*Tal1*, *Gata2*, *Gata3*, *Vsx1*, *Vsx2*, *Lhx4*) were strongly expressed in the post-mitotic precursors at lineage bifurcation point and at the start of the lineage branches (*Figure 1—figure supplement 1A*).

To find gene regulatory elements in the rV2 cell types, we applied scATAC-seq in E12.5 ventral r1 cells. We clustered the r1 scATAC-seq cells based on their open chromatin regions (called features below), followed by integration with the E12.5 scRNA-seq data (*Stuart et al., 2019*) that provides an estimate of the transcript levels for scATAC-seq measured cells. From the scATAC-seq data, we readily identified the cell groups of the rV2 lineage, expressing TFs specific for GABAergic and glutamatergic cell types (*Figure 1—figure supplement 2*). We identified two progenitor cell groups (PRO1 and PRO2), rV2 common precursors (CO), GABAergic neuron branch consisting of four cell groups and glutamatergic branch of two cell groups (*Figure 1—figure supplement 2A*). Further clustering of the isolated rV2 cell groups yielded two groups of common precursors (CO1 and CO2), six cell groups in the GABAergic branch (GA1–GA6), and five cell groups in the glutamatergic branch (GL1–GL5) (*Figure 1D, F*; *Figure 1—figure supplement 2B*). The placement of the scATAC-seq rV2 cells in the pseudotemporal axis (*Figure 1E*) assigned the early state in pseudotime to cells expressing markers of proliferative progenitors and early post-mitotic common precursors, while late pseudotime state

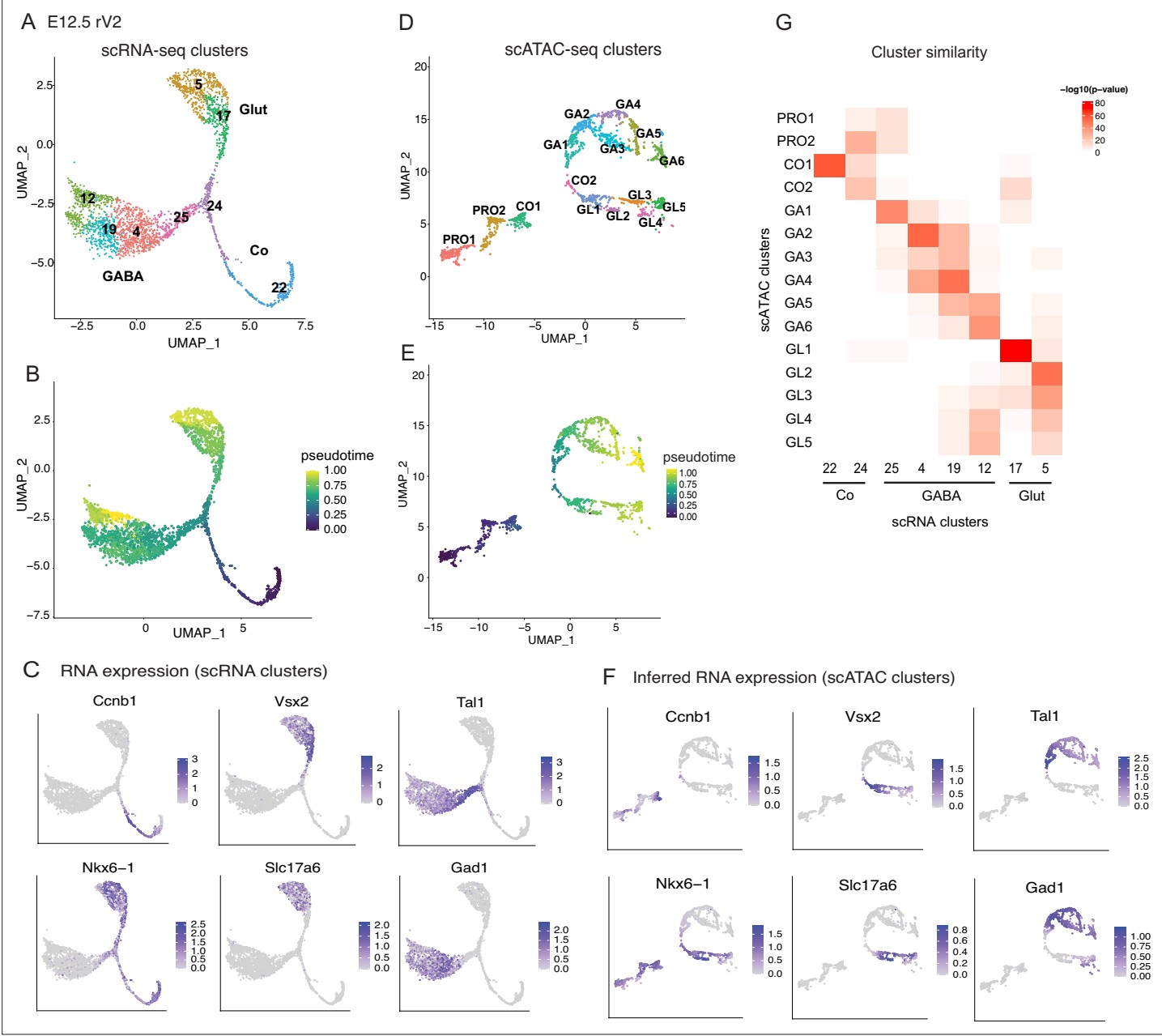

**Figure 1.** Differentiation of the E12.5 mouse rV2 GABAergic and glutamatergic neurons based on transcriptome and chromatin accessibility. (**A**) Uniform Manifold Approximation and Projection (UMAP) of rV2 lineage cells based on scRNA profiles. (**B**) Pseudotime (VIA) scores shown on the scRNA UMAP projection. (**C**) RNA expression of marker genes of progenitors and postmitotic precursors of GABAergic and glutamatergic neurons shown on the scRNA UMAP projection. (**D**) UMAP projection of rV2 lineage cells based on scATAC profiles. (**E**) Pseudotime (VIA) scores shown on the scATAC UMAP projection. (**F**) Inferred RNA expression of the marker genes of progenitors and postmitotic precursors of GABAergic and glutamatergic neurons shown on the scATAC UMAP projection (interpolated expression values). (**G**) Heatmap of scATAC and scRNA cluster similarity scores.

The online version of this article includes the following figure supplement(s) for figure 1:

**Figure supplement 1.** E12.5 mouse rhombomere 1 single-cell RNA-seq.

**Figure supplement 2.** E12.5 mouse rhombomere 1 single-cell ATACseq.

cells express markers of differentiating GABAergic and glutamatergic precursors. The cell groups had consistently increasing average pseudotime from PRO1 to the end of both GABAergic and glutamatergic branches. We next compared the scRNA-seq and scATAC-seq cell group markers across the rV2 lineages (top 25 marker genes per group, similarity testing using hypergeometric distribution)

(*Figure 1G*, *Supplementary file 1*; *Supplementary file 2*). The similarity scores indicate discrete matches between the scATAC-seq and scRNA-seq groups, with the exception of PRO1 and PRO2, as the rV2 scATAC-seq groups expressing progenitor markers (*Nes* and *Phdgh*, *Figure 1—figure supplement 2A*) did not have clear counterparts among the scRNA-seq groups (*Figure 1—figure supplement 1A*) and the progenitors were therefore excluded from the scRNA-seq based lineage analysis. In conclusion, open chromatin features largely reflected the gene expression in the ventral r1 and allowed placing the developing rV2 neurons in cell groups and in a pseudotemporal order.

## Chromatin accessibility profiles in genomic features linked to selector genes in the rV2 cell groups

*Tal1*, *Gata2*, and *Gata3* are required for GABAergic fate selection in rV2 and their expression is activated in the GABAergic post-mitotic precursors, whereas *Vsx2* is expressed in the glutamatergic precursors soon after rV2 lineage bifurcation (*Figure 1—figure supplement 1B*, *Figure 1—figure supplement 2B*; *Achim et al., 2013*; *Lahti et al., 2016*; *Morello et al., 2020a*). To understand the mechanisms behind activation of the expression of these lineage-specific TFs, we analysed the accessibility of chromatin features in *Tal1*, *Gata2*, *Gata3*, and *Vsx2* loci in the rV2 cell groups. To find selector gene-associated chromatin features, we tested all features within the topologically associating domain (TAD, *Singh and Berger, 2021*) containing the selector gene transcription start site (TSS) for correlation between the feature accessibility and the selector gene expression. We defined the features linked to the selector gene with LinkPeaks (*Stuart et al., 2021*) z-score below –2 or above 2 and p-value <0.05 as candidate cis-regulatory elements (cCREs) of the gene (*Supplementary file 3*).

The *Tal1* locus, ±50 kb of the TSS, contains genes *Stil*, *Tal1*, *Pdzk1ip1*, and *Cyp4x1* and 20 scATAC-seq features (*Figure 2A*). In total, the *Tal1* TAD contains 107 features. Accessibility of 15 out of the 107 features in the TAD containing the *Tal1* gene was linked to the *Tal1* expression (p-value <0.05) (*Supplementary file 3*, *Tal1* cCREs). For example, the gain in the accessibility of *Tal1* cCREs at +0.7 and +40 kb correlated temporally with the up-regulation of *Tal1 mRNA* levels, strongly increasing in the earliest GABAergic precursors (GA1) and maintained at a lower level in the more mature GABAergic precursor groups (GA2–GA6), as well as *Pdzk1ip1* mRNA expression, increasing transiently in the earliest GABAergic precursors (GA1) (*Figure 2A, B*). Other features within the *Tal1* gene (+7 kb) as well as in more distal intergenic area (+15.1, +16, +16.4, and +23 kb cCREs) were also found to correlate with the *Tal1* mRNA expression (*Figure 2A, B*, *Supplementary file 3*). Notably, the identified *Tal1* promoter feature (*Tal1* +0.7 kb cCRE) contains *Tal1* midbrain and hindbrain-spinal cord enhancers characterized earlier (*Bockamp et al., 1997*; *Sinclair et al., 1999*). The *Tal1* +40 kb cCRE corresponds to the previously characterized enhancer region driving expression in the midbrain and hindbrain (*Ogilvy et al., 2007*). Enhancers at –3.8 and +15.1/16/16.4 kb of *Tal1* have been shown to drive *Tal1* expression in haematopoietic stem cells and erythrocytes (*Göttgens et al., 2004*; *Supplementary file 4*, *Figure 2—figure supplement 1A*).

Similarly to the *Tal1* locus, the TADs of *Gata2* and *Gata3* genes contained features where accessibility increased specifically in the GABAergic precursors, correlating with the activation of *Gata2* and *Gata3* mRNA expression (*Figure 2—figure supplement 2*, *Supplementary file 3*). Of these, two *Gata2* cCREs, at –3.7 and +0 kb, are located in the proximal regulatory and promoter regions, which have been earlier shown to function as *Gata2* mid- and hindbrain expression enhancers (*Nozawa et al., 2009*), as well as critical *Gata2* autoregulation elements in haematopoietic cells (*Kobayashi-Osaki et al., 2005*; *Supplementary file 4*, *Figure 2—figure supplement 1B*). We also identified an intronic *Gata2* cCRE at +11.6 kb that may mediate the regulation of *Gata2* expression in the spinal cord V2 interneurons by Gata2–Tal1 co-binding (*Joshi et al., 2009*). Four additional previously uncharacterized *Gata2* cCREs were identified downstream of the *Gata2* gene (+22, +25.3, +32.8, and 47.5 kb cCREs, *Figure 2—figure supplement 2A*) and five more further downstream within the TAD (*Figure 2—figure supplement 1B*, *Figure 2—figure supplement 2B*, *Supplementary file 3*). The *Gata2* +22, +25.3, and +32.8 kb cCREs were accessible in the early precursors, before the opening of the *Gata2* –3.7 kb cCRE in GA1 and *Gata2* +0 kb cCRE in GA3–GA6 (*Figure 2—figure supplement 2B*, *Supplementary file 3*).

At the *Gata3* locus, the GABAergic precursor-specific features included several intragenic features (*Gata3* +11.1, +17.2, and +18.8 kb cCREs), and five features upstream the gene at –2.7 to –7.7 kb of *Gata3* TSS (*Supplementary file 3*, *Figure 2—figure supplement 2C, D*). The Gata3 +0.6 and +2.2 kb

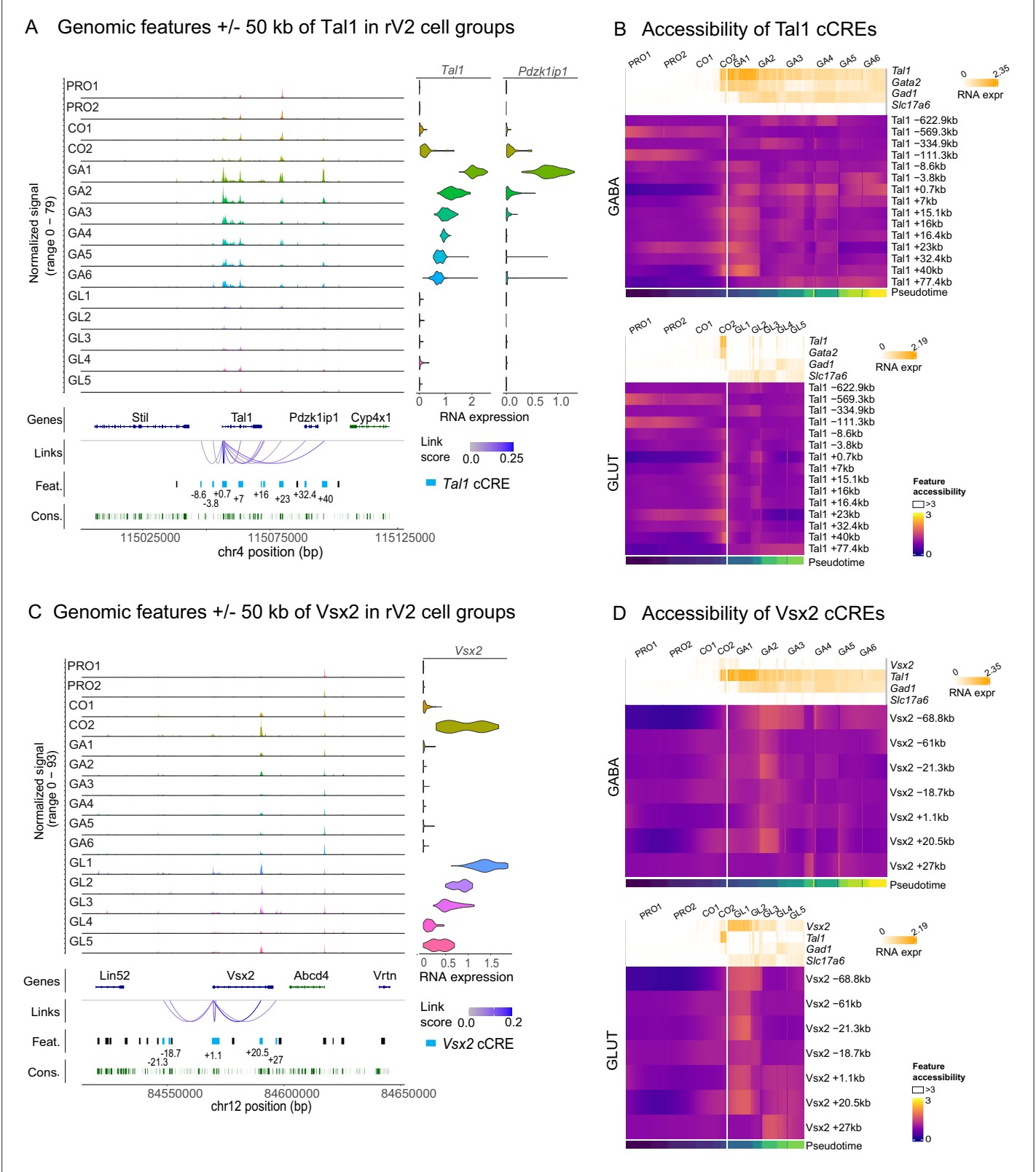

**Figure 2.** Accessibility of chromatin features in the *Tal1* and *Vsx2* loci during GABAergic and glutamatergic development. (**A**) Chromatin accessibility per cell group (normalized signal), Ensembl gene track (Genes), scATAC-seq features (Feat.), feature linkage to gene (Links with the LinkPeaks abs(zscore) >2) and nucleotide conservation (Cons.) within ±50 kbp region around *Tal1* transcription start site (TSS). Violin plots on the right show the expression levels of *Tal1* and *Pdzk1ip1* mRNA per the scATAC-seq cell group (indicated on the left). (**B**) Spline-smoothed *z*-score transformed heatmaps

*Figure 2 continued on next page*

*Figure 2 continued*

of chromatin accessibility at *Tal1*-linked scATAC-seq features (*Tal1* cCREs) in the single cells of rV2 GABAergic and glutamatergic lineages with RNA expression of *Tal1*, *Gata2*, *Gad1*, and *Slc17a6* (sliding window mean(width = 6) smoothed) as column covariable. Cells on the x-axis are first grouped per cell group (top) and then ordered by the pseudotime (bottom) within each group. (**C**) Same as in A, but for the *Vsx2* locus. (**D**) Accessibility of *Vsx2* cCREs in the rV2 GABAergic and glutamatergic lineages, shown as in (**B**). The expression levels of *Vsx2*, *Tal1*, *Gad1*, and *Slc17a6* are shown above the heatmaps.

The online version of this article includes the following figure supplement(s) for figure 2:

**Figure supplement 1.** Comparison of the previously characterized enhancers of *Tal1*, *Gata2*, *Gata3*, and *Vsx2* with the scATAC features identified in this study.

**Figure supplement 2.** Genomic features and feature accessibility at *Gata2* and *Gata3* gene loci.

**Figure supplement 3.** Comparison of the previously characterized enhancers of *Tal1*, *Gata2*, *Gata3*, and *Vsx2* with the scATAC features identified in this study.

and the five upstream cCREs locate within the previously identified Gata3 CNS enhancer region (*Lakshmanan et al., 1999*), and two of those also match *Gata3* lens expression enhancers (*Martynova et al., 2018*; *Supplementary file 4*, *Figure 2—figure supplement 1C*). The *Gata3* +11.1 kb cCRE corresponds to the *Gata3* proximal enhancer driving expression in PNS regions (*Lieuw et al., 1997*). The *Gata3* +17.2 kb cCRE is located in the introns 3 and 4 of the *Gata3* gene and may have enhancer activity in spinal cord V2b interneurons (*Joshi et al., 2009*; *Supplementary file 4*, *Figure 2—figure supplement 1C*).

Importantly, the accessibility of the cCREs of the GABAergic selector TF genes changed with different kinetics during neuronal differentiation. For example, the accessibility of *Tal1* +15.1, +16, +23, and *Tal1* +40 kb cCREs was robustly detected before and at the onset of GABAergic differentiation and declined thereafter (*Figure 2B*). The *Tal1* +23 kb cCRE contained two scATAC-seq peaks, having temporally different patterns of accessibility. The feature is accessible at the 3′ position early and gains accessibility at 5′ positions concomitant with GABAergic differentiation (*Figure 2A*). Detailed feature analysis indicated that the 3′ end of this feature contains binding sites of *Nkx6-1* and *Ascl1* that are expressed in the rV2 neuronal progenitors, while the 5′ end contains TF-binding sites of Insm1 and Tal1 TFs that are activated in early precursors (described below, see *Figure 3C*). The accessibility of the Tal1 +0.7 kb cCRE largely reflected the *Tal1* mRNA expression and, unlike the early-peaking cCREs, such as *Tal1* +40 kb, it was accessible also in the late GABAergic neuron precursors GA3–GA6 (*Figure 2B*). Thus, distinct regulatory elements of the selector TFs may be responsible for the activation and for the maintenance of the lineage-specific expression of selector TF gene.

*Vsx2* gene expression was detected in the common precursors CO1 and increased in CO2 and the glutamatergic precursors (GL1–GL5) (*Figure 2C*, RNA expression). At the *Vsx2* locus, an intronic feature (*Vsx2* +20.5 kb) and the distal features at –61 and –21.3 kb of *Vsx2* TSS were opened concomitant with its expression activation, whereas accessibility of the *Vsx2* promoter region (*Vsx2* +1.1 kb cCRE) and +27 kb cCRE increased later in the glutamatergic precursors (*Figure 2D*).

The information about the enhancers of *Vsx2* brain expression is scarce, but our *Vsx2* r1 cCREs at –18.7, –21.3, +1.1, and +20.5 kb match previously characterized enhancers of *Vsx2* expression in photoreceptor cells and retinal bipolar neurons and also contain putative autoregulatory *Vsx2*-binding sites occupied in developing spinal cord motor neuron and interneuron lineages (*Supplementary file 4*, *Figure 2—figure supplement 1D*).

Developmentally important gene regulatory elements are often conserved. We therefore asked if features detected by scATAC-seq could present more evolutionary conserved sequences than random sequences of mouse genome. To that aim, we used nucleotide conservation scores from UCSC (*Siepel et al., 2005*). We overlaid conservation information and scATAC-seq features to both validate feature definition as well as to provide corroborating evidence to recognize cCRE elements. We calculated statistical enrichment of conserved nucleotides within exons, introns and our scATAC-seq features (*Supplementary file 5*, sheet per chromosome). Systematically, odds ratios per chromosome were highest for exons and lowest for introns, odds ratios for features were between these two. All of the odds ratios showed a significant difference from what is expected if there was no relationship between the nucleotides within features and conservation ($p = {<}2.20e{-}16$, Fisher's exact test, *Supplementary file 5*, sheet per chromosome). The conservation of chromatin features is consistent with a putative role as regulatory sites.

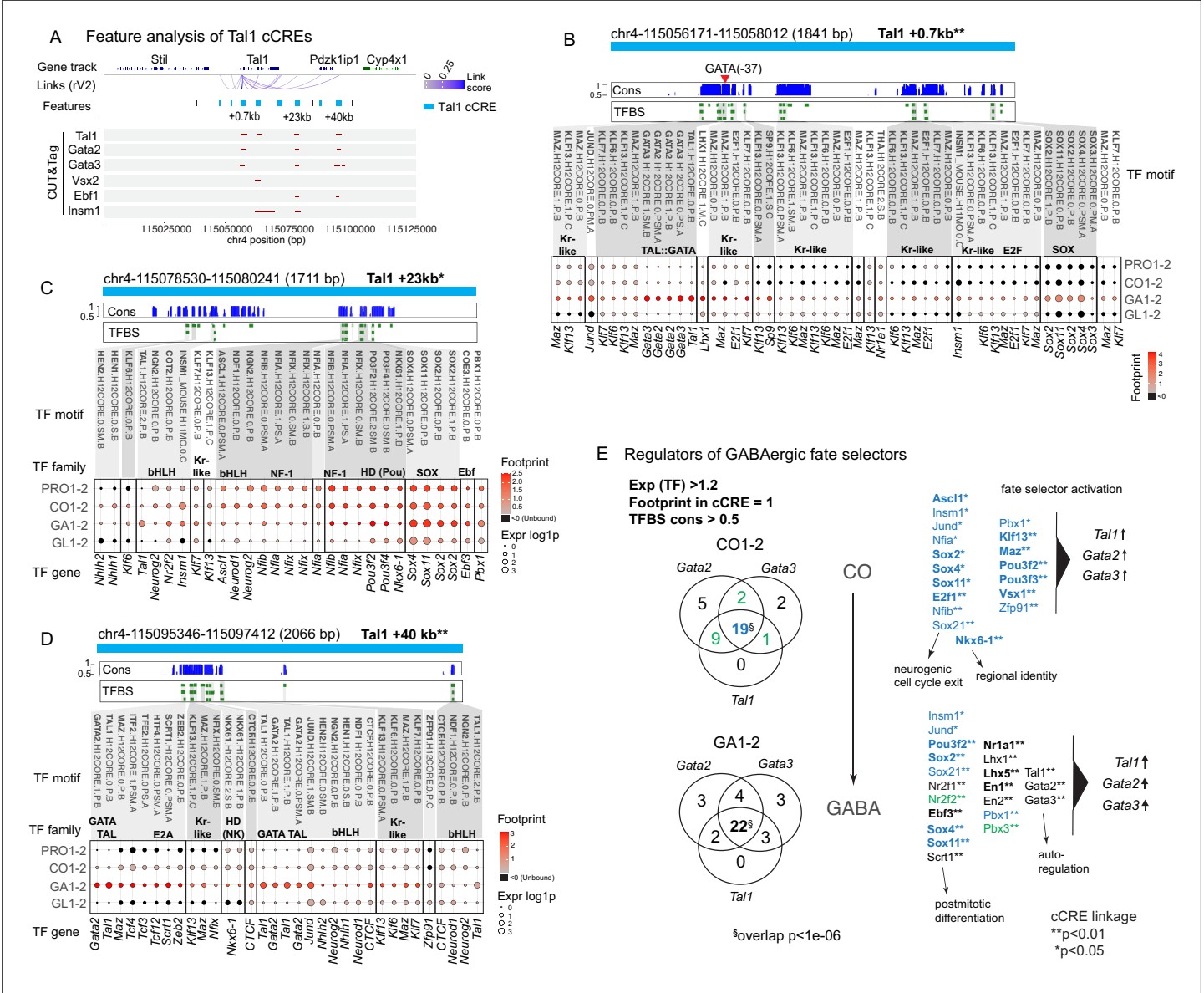

**Figure 3.** Transcription factor (TF) binding in putative regulatory elements of *Tal1* and the overlap between TFs interacting with other GABAergic fate selector genes. (**A**) scATAC features within ±50 kb of the *Tal1* gene. *Tal1* cCREs are shown in blue. CUT&Tag: CUT&Tag consensus peaks indicating Tal1, Gata2, Gata3, Vsx2, Ebf1, and Insm1 binding in the E12.5 mouse r1. No Tead2 CUT&Tag consensus peaks were located in this region. (**B–D**) Footprint analysis of the features at +1, +23, and +40 kb of the *Tal1* transcription start site (TSS). Footprint scores at conserved TFBSs are shown for progenitors (PRO1–2), common precursors (CO1–2), GABAergic precursors (GA1–2), and glutamatergic precursors (GL1–2). In each dot plot, the strength of footprint at TFBS in the feature is shown as colour (Footprint score, average of cell group) and the expression of the TF gene in dot size (log1p). TFBS names (Hocomoco v12) are shown at the top and the TF gene names (mouse) are shown under the dotplot. The red arrowhead in (**B**) indicates the conserved Gata2 TFBS at –37 bp position required for the neural expression of Tal1 (see also ***Supplementary file 4***). (**E**) Overlap of the TFs with footprints in the cCREs of *Tal1*, *Gata2*, and *Gata3* in the common precursors of rV2 lineages (CO1–2) and in the rV2 GABAergic precursors (GA1–2). Venn diagrams show the number of TFs with an scATAC footprint at conserved TFBS in *Tal1*, *Gata2*, or *Gata3* cCREs and with the gene expression (log1p) >1.2 in the analysed cell group (Exp(TF) >1.2; Footprint in cCRE = 1; TFBS cons >0.5). The TFs associated with the cCREs of all three selector genes in common precursors (CO1–2) and GABAergic precursors (GA1–2) are listed. Blue text: 19 TFs that interact with cCREs of *Tal1*, *Gata2*, and *Gata3* in the CO1–2 cells. Some of these TFs continue to be expressed and interact with the *Tal1*, *Gata2*, and *Gata3* genes in the GA1–2 cells. Twenty-two TFs interact with *Tal1*, *Gata2*, and *Gata3* in GA1–2 cells. Green text, TFs found associated with two selector genes in CO1–2 (green in the Venn diagram of CO1–2) and associated with all three selector genes in the GA1–2. Black text, TFs co-regulating the selector TFs in GA1–2 and not expressed in the CO1–2 cells. The TFs regulating both GABAergic and glutamatergic selectors are marked in bold. § The probability of finding n overlapping genes considering all mouse genes equally is p < 1e–6. *,** The collective minimum statistical significance of feature to gene links for selector genes *Tal1*, *Gata2*, and *Gata3* cCREs for the given TF is shown as: *p-value <0.05; **p-value <0.01 (with LinkPeaks z-score above 2 or below –2).

*Figure 3 continued on next page*

*Figure 3 continued*
The online version of this article includes the following figure supplement(s) for figure 3:

**Figure supplement 1.** Transcription factor (TF) signatures found by scATAC footprinting.

**Figure supplement 2.** Regulatory features of the glutamatergic fate selector *Vsx2* and common regulators of *Vsx2* and *Vsx1*.

**Figure supplement 3.** CUT&Tag analysis of Tal1- and Vsx2-associated chromatin in the E12.5 mouse r1.

**Figure supplement 4.** CUT&Tag analysis of Gata2- and Gata3-associated chromatin in the E12.5 mouse r1.

**Figure supplement 5.** CUT&Tag analysis of Ebf1-, Insm1-, and Tead2-associated chromatin in the E12.5 mouse r1.

**Figure supplement 6.** Footprint position density over length-normalized CUT&Tag peaks per each transcription factor (TF).

In summary, we identified the genomic features that are linked to GABAergic and glutamatergic fate selector genes, and where chromatin accessibility changes specifically during rV2 neuron differentiation. Those features may represent regulatory elements of the selector genes. Furthermore, literature research revealed that the previously characterized enhancers, and especially the nervous system expression enhancers of *Tal1*, *Gata2*, *Gata3*, and *Vsx2* mostly are found within the ATAC features identified here (63.6–100% of gene enhancers overlapping with scATAC-seq features, *Figure 2—figure supplement 3B*).

## TF activity at cCREs of the rV2 cell fate selector genes

Next, we wanted to identify the TFs associated with the putative regulatory elements of *Tal1*, *Gata2*, *Gata3*, and *Vsx2* genes. TF binding to chromatin affects its accessibility, leaving a footprint which can be detected in scATAC-seq data (*Bentsen et al., 2020*). We applied the footprint analysis across the rV2 cell groups using the known TF-binding sites (TFBS) from the HOCOMOCO collection (*Vorontsov et al., 2024*). The TF footprinting analysis identified global TF activity patterns correlating with GABA- and glutamatergic neuron differentiation (*Figure 3*, *Figure 3—figure supplement 1*). To find the TFs that may regulate the expression of *Tal1*, *Gata2*, *Gata3*, and *Vsx2*, we identified the TFBSs at the conserved genomic locations (weighted average conservation at TFBS >0.5) within each selector gene cCRE. The results of the footprint analysis are shown for selected features of *Tal1* (*Figure 3A–D*) and *Vsx2* (*Figure 3—figure supplement 2A, C–E*), showing only the TFs expressed (average RNA expression >1.2 (log1p)) in any of the four cell groups from PRO1–2, CO1–2, GA1–2, and GL1–2. We observed both common and distinct patterns of TF binding in the cCREs of *Tal1* and *Vsx2* in the rV2 common precursors and early GABA- and glutamatergic cells. Both *Tal1* and *Vsx2* cCREs contain conserved binding sites of Kr-like TFs, bHLH TFs, E2A, Ebf, E2F, NF-1, Pbx, and Sox TF families, and various HD TFs (*Figure 3B–D*, *Figure 3—figure supplement 2C–E*). The analysis of TFBSs at the footprint regions suggested that each region may have affinity to several different TFs belonging to a TF class. For example, four Sox TFs are expressed in early rV2 lineage cells (*Sox2*, *Sox3*, *Sox4*, and *Sox11*), which all may compete for the same binding sites in *Tal1* +0.7 kb cCRE or *Vsx2* –61 kb cCRE (*Figure 3C*, *Figure 3—figure supplement 2D*).

To further understand the mechanisms of the activation of lineage-specific selector expression, we asked which TFs commonly and specifically regulate the GABAergic selector TF genes *Tal1*, *Gata2*, and *Gata3*. Identification of the common regulators of GABAergic fate selectors revealed 21 TFs in CO1–2 and 22 TFs in the GA1–2 cell groups (*Figure 3E*). Interestingly, the common regulators between the stages are overlapping partially, with the TFs associated with the neurogenic cell cycle exit predominant in the common precursors (CO) and the postmitotic differentiation and the autoregulation network TFs showing activity later in the GABAergic precursors.

We next asked which upstream regulators are common to both rV2 GABAergic (*Tal1*, *Gata2*, and *Gata3*) and glutamatergic (*Vsx1* and *Vsx2*) fate selector genes. We identified 12 TFs, including Ascl1, E2f1, Nkx6-1, and Vsx1 that may regulate both GABA- and glutamatergic selectors in the CO1–2 cell group (*Figure 3E*, *Figure 3—figure supplement 2F*, genes indicated in bold). The distinct regulators of GABA- and glutamatergic fate selectors, aside from the lineage-specific autoregulatory functions of the selectors themselves, include Lhx1 and En2 in the GABAergic sublineage (Lhx1-binding *Tal1* +0.7 kb feature, *Figure 3B*), and Lhx4, Shox2, and Nhlh2 regulating *Vsx2* expression in the glutamatergic lineage (*Figure 3—figure supplement 2C-E*). Footprints at Nkx6-1 TFBS were detected in the cCREs of all selectors in common precursors, but after onset of differentiation, they become restricted to the cCREs of *Vsx2* and *Vsx1* in the glutamatergic cells. Interestingly, the *Vsx2* –68.8 kb

cCRE contains conserved GATA TF-binding sites (*Figure 3—figure supplement 2E*). Conversely, Vsx1- and Vsx2-binding sites were found in the cCREs of *Gata2* and *Gata3*, suggesting a mutual cross-regulation between the GABA- and glutamatergic selector genes. Putative Vsx1-binding sites were detected in all three GABAergic fate selector genes, *Tal1*, *Gata2*, and *Gata3*, in the rV2 common precursors (*Figure 3E*, CO, CO1–2).

The proneural bHLH TFs appeared to contribute to the activation of the selector genes. Ascl1 and Neurog2 TFBS are found at the cCREs of both *Tal1* (*Figure 3C, D*) and *Vsx2* (*Figure 3—figure supplement 2C, D*). In addition, Neurog1 TFBS was bound at *Vsx2* +20.5 and –61 kb cCREs in the rV2 common precursors (*Figure 3—figure supplement 2C, D*). Interestingly, several cell cycle-associated TFs may also contribute to the regulation of selector gene expression. Those include Sox2, Sox21, Insm1, Ebf1, Ebf3, and E2f1. E2f1, and Ebf1–3-binding sites are located within the GC-rich areas (*Figure 3B–D*, *Figure 3—figure supplement 2C–E*). E2f1 binds Tal1 cCREs +0.7 and +7 kb and is strongly expressed in the CO1–2, the common precursors of rV2 GABA- and glutamatergic cells (*Figure 3B*). The expression of *E2f1* declines after differentiation, while *Ebf1–3* are expressed in common precursors (CO1–2) and maintained in differentiating neurons of both GABA- and glutamatergic sublineages (GA1–2 and GL1–2) (*Figure 3B–D*, *Figure 3—figure supplement 2C–E*). Interestingly, the scATAC-seq feature analysis suggested that Ebf1–3 TFs interact with Tal1 cCREs as well as Gata2 cCREs in both GABA- and glutamatergic cells (*Supplementary file 3*).

In addition to the footprint analysis, we analysed chromatin interactions of Tal1, Gata2, Gata3, Vsx2, Ebf1, and Insm1 in the E12.5 r1 by CUT&Tag (*Figure 3—figure supplement 3*, *Figure 3—figure supplement 4*, *Figure 3—figure supplement 5*). The binding of Tal1, Gata2, and Gata3 at the *Tal1* cCREs was supported by CUT&Tag (*Figure 3*, *Figure 3—figure supplements 3 and 4*), consistent with the autoregulatory activity of the GABAergic fate selector genes. In addition to the *Tal1* +0.7 and +40 kb cCREs, we detected Tal1, Gata2, and Gata3 binding at the *Tal1* +23 kb cCRE, where scATAC-footprinting did not indicate binding at conserved sites (*Figure 3A, C*). The DNA binding of Insm1 and Ebf1 was also partially confirmed by CUT&Tag. Insm1 footprints were found in Tal1 cCREs –3.8 and +23 kb, while Insm1 CUT&Tag peaks were found at Tal1 +7, and +23 kb cCREs (*Figure 3A*, CUT&Tag Insm1). scATAC-footprint was found at the conserved Ebf1 TFBS in *Tal1* +77.4 kb cCRE and, by CUT&Tag, Ebf1 was found to interact with *Tal1* +23 kb and *Tal1* +40 kb cCREs (*Figure 3A*). However, scATAC-seq footprints for Ebf1 are found also in the *Tal1* +23 kb and *Tal1* +40 kb cCREs, whereas those TFBS are weakly conserved across mammals (cons <0.5). Overall comparison of TF binding detected by scATAC-seq footprinting and by CUT&Tag indicates strong bias towards the same features (*Supplementary file 6*). Earlier reports of corroborations between Cut&Run and footprinting have shown 9% (*Lmx1b*) and 30.5% (*Pet1*) of footprints having overlapping Cut&Run peak (*Eastman et al., 2025*). In our analysis, equivalent percentages range from 5.3% to 25.9%, depending on the TF. Additionally, an analysis of footprint positions in relation to CUT&Tag peaks indicates clear tendency of having most footprints towards the centre of the peaks (*Figure 3—figure supplement 6*), further validating eligibility of CUT&Tag and footprinting as corroborating methods.

In summary, the regulatory feature analysis revealed several mechanisms that may contribute to the timing of activation and the maintenance of selector TF expression. Firstly, we found the binding sites of cell cycle progression or progenitor identity-associated TFs E2f1 and Insm1, proneural TFs Ascl1, Neurog1, Neurog2, Neurod1, and Neurod2, as well as TFs expressed after cell cycle exit, such as Sox4 and Ebf3 at the putative distal regulatory elements. Conserved binding sites for other Ebf TFs, Ebf1 and Ebf2, are also found in the cCREs of *Tal1* and *Gata2*, but not *Gata3*. In addition, the proximal elements and the promoter regions of *Gata2* and *Tal1* genes contained footprints at conserved GATA–TAL motif in the GABAergic neurons, indicating autoregulation of the selector TFs. Similarly, conserved Vsx1 and Vsx2 motifs in *Vsx2* cCREs were associated with scATAC-seq footprints in the glutamatergic neurons.

## Pattern of expression of the putative regulators of Tal1 in the r1

Our analysis of TFBSs in the selector gene cCREs indicated that GABAergic selector TF expression may be regulated by tens of TFs. From these factors, we chose a subset of TFs (*Insm1*, *Sox4*, *E2f1*, and *Ebf1*) for expression studies based on earlier knowledge of their functions. Among those, E2f1 and Ebf TFs are involved in cell cycle progression and neuronal differentiation. Previous in situ hybridization data (Allen Brain Atlas, developing mouse brain) indicates that of *Ebf1–3*, *Ebf1* and *Ebf3* are

expressed in the developing r1 at E11.5 and E13.5. Here, we chose to investigate the expression pattern of *Ebf1* and its overlap with *Tal1* more closely. Our analysis also indicated several Sox TF family proteins of Sox B and C class. The roles of SoxB1 and SoxB2 proteins Sox2 and Sox21 have been well studied in early neurogenesis (*Makrides et al., 2018*; *Sandberg et al., 2005*). Interestingly, we found that SoxC class TFs Sox4 and Sox11 may be regulating *Tal1* expression in the early rV2 GABAergic neurons (*Figure 3C* Tal1 +23 kb cCRE, Sox4).

In addition, we found a mediator of Hippo signalling pathway, Tead2, among the possible regulators of *Tal1*. The regulation of a selector gene by Tead2 is of interest because of its role as a mediator of mechanical signals from cells that could serve as cues to the apical–basal migration, a process concomitant to the cell cycle exit and fate commitment in neuronal precursors. An scATAC-seq footprint at Tead2 TFBS was found in *Tal1* +0.7 kb cCRE in CO1–2, and interestingly, in a distal *Tal1* cCRE at –662.9 kb; although the conservation scores at those sites were below 0.5.

To test whether the *Insm1*, *Sox4*, *E2f1*, *Ebf1*, and *Tead2* genes are expressed in the *Tal1*+ lineage precursors, we analysed the pattern of their mRNA expression in the E12.5 mouse r1 by mRNA in situ hybridization with RNAscope probes (*Figure 4*). In the embryonic brain, the progression of neuronal differentiation follows the apical–basal axis, as the newly born precursors exit the ventricular zone and migrate to the mantle zone (*Figure 4D*, arrow). *Tal1* is expressed shortly after the cell cycle exit. The expression of *Insm1*, *Sox4*, *E2f1*, *Ebf1*, and *Tead2* mRNA was detected in the ventricular zone of the ventrolateral r1 at E12.5 and overlapped with *Tal1* mRNA expression in the cells located close to the ventricular zone (*Figure 4A–C*, arrowheads in a4, a7, b4, b7, c4). We quantified the intensity of the fluorescence signal of each RNAscope probe along the apical-to-basal axis of the neuroepithelium (*Figure 4D–F*, arrow), and found that the maximum intensity of *Insm1*, *E2f1*, and *Tead2* probe signal is detected slightly closer to the ventricular surface (VS) than that of *Tal1* (*Figure 4D, E*). The expression of *Ebf1* followed the pattern of expression of *Tal1* closely, but was expressed at slightly higher levels earlier (*Figure 4F*). The maximum intensity of *Sox4* mRNA (*Figure 4A, a5–a7, and D*) was detected in the cells at a more basal position in the neuroepithelium, where the differentiated neurons reside. The expression patterns observed along the apical–basal axis in the E12.5 r1 neuroepithelium (*Figure 4D–F*) matched with the expression patterns detected by scRNA-seq in the respective cell groups along the pseudotemporal axis (*Figure 4G–I*).

In conclusion, *Insm1*, *Sox4*, *E2f1*, *Ebf1*, and *Tead2* are expressed in the early rV2 precursors, including the *Tal1*-expressing GABAergic neuron precursors. The expression of these TFs precedes the expression of *Tal1* on the differentiation axis, consistent with a putative role in the initiation of *Tal1* gene expression.

## Regulatory signature of developing rV2 GABAergic neurons

In addition to the regulatory elements of GABAergic fate selectors, we wanted to understand the genome-wide TF activity during rV2 neuron differentiation. To this aim, we applied ChromVAR (*Schep et al., 2017*). Consistent with the footprint analysis in cCREs, ChromVAR analysis indicated that accessible genomic regions in rV2 progenitor and precursor cells are enriched in Sox, NeuroD1 and Nfia/b/x TF-binding motifs (*Figure 5A*). Those TFs have been suggested to regulate temporal dynamics of gene expression in mouse neural tube (*Sagner et al., 2021*; *Zhang et al., 2024*) and the cell cycle exit of early dopaminergic neuron progenitors (*Lahti et al., 2025*). In addition, we found Nhlh1/2, NeuroD2, Bhlhe22, Ebf1–3, Pou3f4, and Lhx5 motifs are accessible in common precursors, shortly preceding the gain of accessibility in Gata2/3 and Tal1 motifs in GABAergic neurons (*Figure 5A*, square). In conclusion, many of the TFs with footprints in selector gene cCREs had a genome-wide activity pattern consistent with a regulatory function.

We also analysed the activity of Tal1, Gata2, and Gata3 in the rV2 neurons more closely. For those TFs, the accessibility at most of their TFBSs closely follows the TF RNA expression (*Figure 5B, C*). Interestingly, there are three TFBS motifs of *Tal1* in the HOCOMOCO v12 resource. Of these Tal1 TFBSs, the activity pattern of the TAL1.H12CORE.0.P.B motif (*Figure 5C*, TAL1.H12CORE.0.P.B) was most specific to the GABAergic lineage, similar to the Gata2 and Gata3 TFBSs (*Figure 5C*). The TAL1.H12CORE.0.P.B is a 19-bp motif that consists of an E-box and GATA sequence and is likely bound by heteromeric Gata2–Tal1 TF complex, but may also be bound by Gata2, Gata3, or Tal1 TFs separately. The other TFBSs of Tal1 contain a strong E-box motif and showed either a lower activity (TAL1.

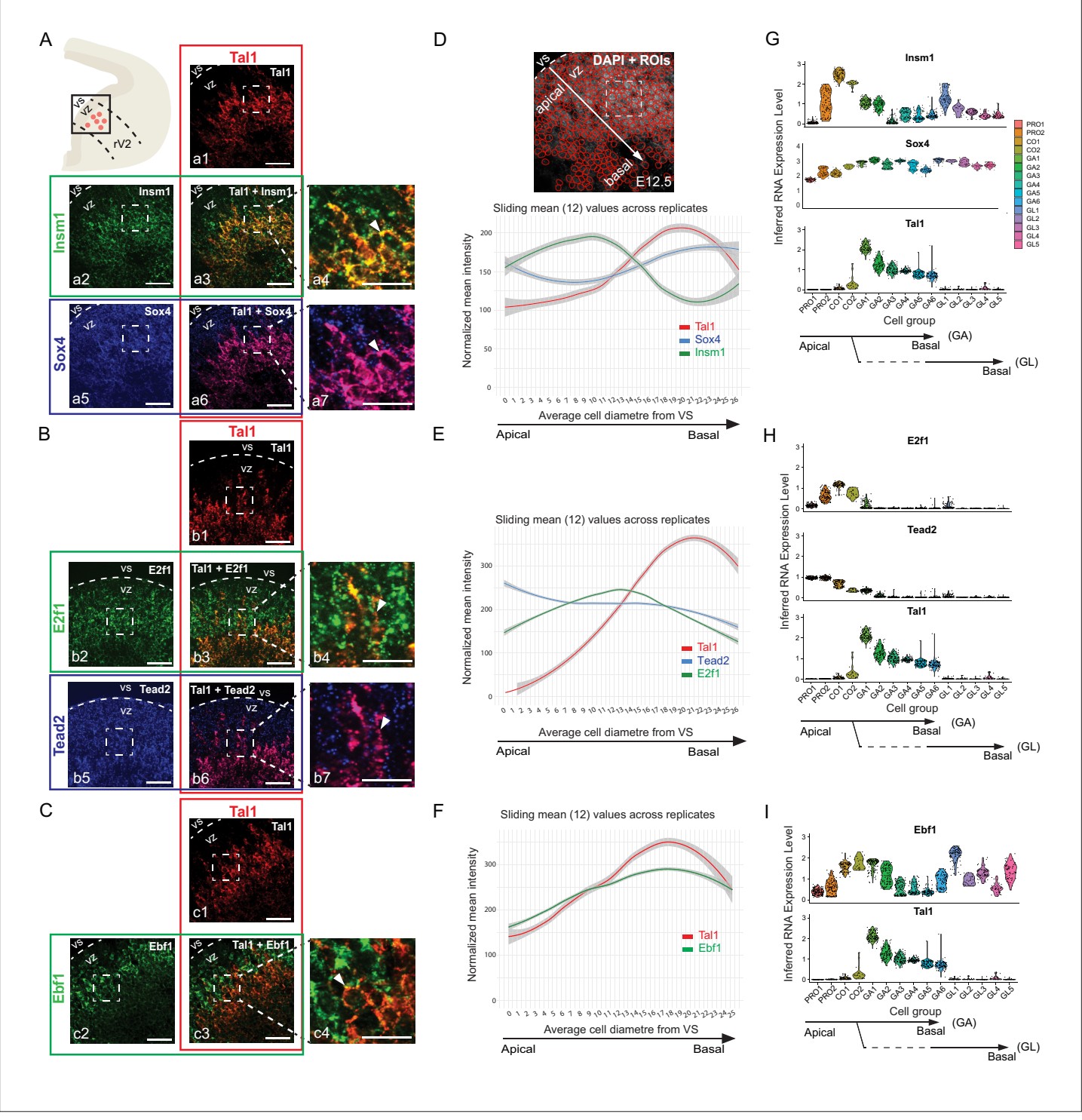

**Figure 4.** Expression patterns of transcription factors (TFs) associated with *Tal1* candidate cis-regulatory elements (cCREs) in the developing anterior brainstem. (**A–C**) mRNA in situ hybridization with the indicated probes on transverse paraffin sections of E12.5 wild-type mouse embryos. A scheme (top left) illustrates the rV2 domain; the boxed area corresponds to the region shown in the images. Dashed lines indicate the ventricular surface (VS), marking the apical border of the ventricular zone (VZ). Dashed boxed areas indicate regions shown in the zoom-in panels. (**A**) mRNA detection of *Tal1* (**a1**), *Insm1* (**a2**), and *Sox4* (**a5**). Overlays show the merged *Tal1/Insm1* (**a3, a4**), and merged *Tal1/Sox4* (**a6, a7**) signal. (**B**) mRNA detection of *Tal1* (**b1**), *E2f1* (**b2**), and *Tead2* (**b5**). Overlays show merged *Tal1/E2f1* (**b3, b4**) and merged *Tal1/Tead2* (**b6, b7**) signal. (**C**) mRNA detection of *Tal1* (**c1**) and *Ebf1* (**c2**), and merged *Tal1/Ebf1* (**c3, c4**) signal. Arrowheads point to the co-localization of probes. The co-localization signal is mostly cytoplasmic. Scale bars: 25 μm in zoom-ins (**a4, a7, b4, b7, c4**), others 50 μm. (**D–F**) Quantification of the mRNA in situ hybridization signal. An example of StarDist

*Figure 4 continued on next page*

Figure 4 continued

regions of interest (ROIs) overlaid with DAPI staining is shown in (**D**). The fluorescence intensity per cell is plotted along the apical–basal differentiation axis. Arrows indicate the apical-to-basal axis. In all plots, the y-axis represents average fluorescence intensity per cell; the x-axis represents the distance from the ventricular surface (VS) using average cell diameter as the unit. Lines represent the sliding mean (12) of three biological replicates as fitted with LOESS (locally estimated scatterplot smoothing) using geom_smooth(method = "loess", se = TRUE, level = 0.95) in ggplot2. The 95% ($\alpha$ = 0.05) confidence interval (the shaded area) is derived from the standard errors of the smoothed values. Number (n) of datapoints for fit and CI are for D (833, 895, 839), E (768, 863, 878), F (851,921) in the order of channels shown in the figure. (**G–I**) Violin plots showing mRNA expression, inferred from scRNA-seq, in the rV2 neuronal populations. The pseudotime order of the GABAergic and glutamatergic lineage cell groups is shown by the branched arrows.

H12CORE.1.P.B) or an earlier peak of activity in common precursors with a decline after differentiation (TAL1.H12CORE.2.P.B) (*Figure 5C*).

## Targets of the GABAergic fate selectors in the developing rhombomere 1

We next focused on the roles of GABAergic fate selector genes in determining the GABA- vs glutamatergic phenotype in rV2 cell lineages. Using the combined evidence of TF footprints, TFBS conservation and in vivo CUT&Tag experiments (*Figure 6A*), we identified 3948 target genes of Tal1, 780 target genes of Gata2, and 721 target genes of Gata3 (*Figure 6B*, *Supplementary file 7*). About half of the identified target genes are differentially expressed in GABA- or glutamatergic neurons (differential expression between the GA1–2 and GL1–2 groups with p-value <0.01 in Wilcoxon's test). However, most of the identified target genes of Tal1, Gata2, and Gata3 are lowly expressed in the GABA- and glutamatergic neurons, with only 30% of the genes being expressed above RNA avg(exp) >0.5 (log1p) in GA1–2 or GL1–2 cell groups (*Supplementary file 7*). Gene set enrichment analysis (GSEA) (*Subramanian et al., 2005*) showed that Tal1 target genes are expressed in both GABA- and glutamatergic neurons (*Figure 6C*), while Gata2 and Gata3 target genes are overexpressed in GABAergic over glutamatergic precursors (*Figure 6H, M*). To further characterize the identified targets, we compared the numbers of target features associated with the target gene by a positive or negative feature-to-gene expression link (*Figure 6D, I, N*). The positive links outnumbered negative links by roughly a factor of 2, which may reflect transcriptional activator function of the studied TFs, but also the complexity of gene regulation as there could be multiple positive and negative links to each gene. We also calculated the amount of target genes considering their closest linked feature with distance bins of 0–5, 5–50, and >50 kb (*Figure 6E, J, O*). The results showed a largely expected distribution, where the category of longest links contained most target genes. This is likely due to the category being limited by TADs, which are often large, yielding considerable amounts of distal regulatory elements for many genes. The calculation of expression variability of the target genes, stratified by the distance to closest selector-bound linked feature, showed that the expression variability in rV2 cell groups was higher for proximal feature-linked Tal1 target genes than for distal feature-linked genes (*Figure 6F, K, P*). Thus, regardless of the large amounts of potential distal regulatory elements for many targets, the target genes with proximal selector TF-bound features (0–5 kb) show the most cell-type-specific patterns. However, a few cell-type-specific genes also appeared to be regulated by distal features, for example at >50 kb from the TSS (*St18*, *Nkx6-1*, *Nhlh1*) or 5–50 kb from TSS (*Zfpm1*, *Lmo1*) in the case of Tal1 targets (*Supplementary file 7*). Finally, we also investigated cell type associations of the highly expressed target genes (exp >0.5 (log1p) in GA1–2 or GL1–2 cell groups) by calculating scores against CellMarker 2024 database (*Chen et al., 2013*; *Hu et al., 2023*), revealing clear enrichment of inhibitory interneuron markers among the Tal1, Gata2, and Gata3 target genes (*Figure 6G, L, Q*).

Finally, we analysed the overlap of the cell-type-specific target gene lists, aiming to find genes regulated by all the GABAergic fate selectors (common target genes) or by any selector TF specifically. No common target genes were found among the glutamatergic neuron-specific genes (*Supplementary file 8*). The genes commonly regulated by Tal1, Gata2, and Gata3 are exclusively overexpressed in the GABAergic lineage and contain genes comprising the autoregulatory and GATA co-factor networks (*Gata2*, *Tal1*, *Zfpm1*, *Lmo1*). The GABAergic phenotype (*Gad1*, *Gad2*, *Slc32a1*) is regulated by Gata2 and Gata3 (*Supplementary file 8*, *Figure 7*, *Figure 7—figure supplement 1*). Tal1 seems to be involved in the activation or modulation of Notch signalling pathway genes (*Pdzk1ip1*, *Hes5*). Interestingly, Tal1 also interacts with genes specific to rV2 glutamatergic cells (*Lhx4*, *Nkx6-1*), possibly repressing their expression in the developing GABAergic precursors. However, as *Tal1* is expressed

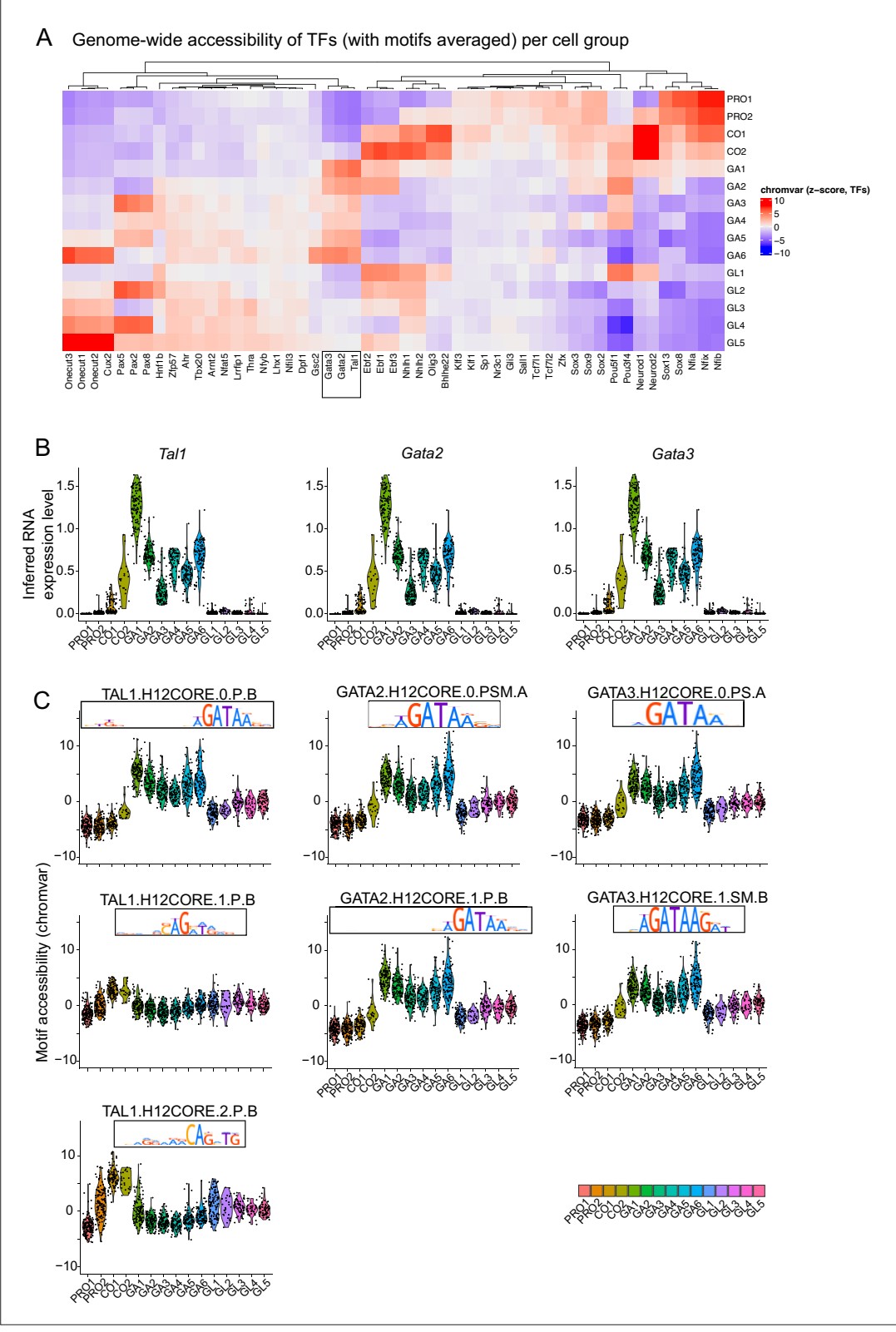

**Figure 5.** Dynamic accessibility of transcription factor (TF)-binding sites suggests a genome-wide function for Gata2, Tal1, and Gata3 in the selection of the GABAergic fate. (**A**) Genome-wide enrichment of TF-binding sites in the accessible regions in GABAergic and glutamatergic lineages. Heatmap of chromVAR z-scores (avg(TFBS motifs per TF)) for the expressed TFs in rV2 cell groups. All TFs, of which a TFBS-motif chromVAR score was among

*Figure 5 continued on next page*

*Figure 5 continued*

top 10 scores in any cell group, are shown. (**B**) Violin plots of the TF gene expression (average scRNA expression) for *Tal1*, *Gata2*, and *Gata3*. (**C**) Motif accessibility (chromVAR score) of Tal1, Gata2, and Gata3 HOCOMOCO v12 TFBS motifs in the rV2 cell clusters. HOCOMOCO v12 contains three Tal1 motifs, two Gata2 motifs, and two Gata3 motifs. Ordering of cell groups is the same in all violin plots.

already in the common precursors of rV2 lineage, the possibility of the activation of *Lhx4* by Tal1 is not excluded. We also suggest the cross-regulation of GABA- and glutamatergic selector genes in the developing rV2 neuronal precursors: Gata3 may repress the *Vsx2* expression in the GABAergic cells, while Vsx2 may repress *Lmo1* in glutamatergic cells, establishing the initiated cell type-specific gene expression programmes (*Figure 7*).

## Discussion

Extensive studies during the past decades have identified several TFs instrumental for neuronal fate decisions in the embryonic brain. How these TFs are regulated and how they guide neuronal differentiation remains less understood. Differentiation of GABAergic and glutamatergic neurons in the ventrolateral r1 is a good model system for TF-guided cell differentiation, as in this region the activation of Tal and Gata TFs is both spatially and temporally controlled, and these TFs guide a binary selection between the two alternative cell fates. In this work, we integrated analyses of gene expression, chromatin accessibility and TF occupancy during bifurcation of GABAergic and glutamatergic lineages in the ventrolateral r1. Our results suggest shared mechanisms connecting selector TF activation to developmental processes and demonstrate combinatory functions of Tal1, Gata2, and Gata3 TFs in the GABAergic fate selection.

### Mechanisms behind selector TF activation

The HD TF code, established by intercellular communication in the neuronal progenitors, defines the competence of differentiation into specific lineages. In a study of the regulation of cortical neuron development, the binding activities of known cortical patterning genes (*Pax6*, *Emx2*, *Nr2f1*) suggested that these patterning TFs function in a combinatorial and synergistic manner and indeed interact with enhancers of many target genes, including the cortical glutamatergic neuron fate selector gene *Fezf2* (*Ypsilanti et al., 2021*). In ventrolateral r1, the Tal1, Gata2, Gata3, and Vsx2 TFs are activated in the region where the proliferative progenitor cells express the HD TF Nkx6-1 (*Achim et al., 2012*; *Lahti et al., 2016*). Our result suggests that Nkx6-1 directly regulates the spatial expression pattern of the selector TFs important for acquisition of GABAergic and glutamatergic neuron phenotypes. Nkx6-1 is not sufficient for selector TF expression, which is activated only when the proliferative progenitor cells exit the cell cycle and give rise to postmitotic precursors. Our results suggest that TFs E2f1, Ebf1, and Insm1 temporally control the selector TFs and connect their expression to cell cycle exit. The expression of *E2f1*, *Ebf1*, and *Insm1* mRNA is not spatially restricted, but is upregulated in early postmitotic precursors across the whole r1 neuroepithelium. Our results show that the expression of these TFs precedes activation of *Tal1*. E2f1 is best known for its function as a positive regulator of cell cycle progression. However, there also is evidence for a more complex role for E2f1 in cell cycle regulation. It has been suggested that the effect of E2f1 is dependent on its level of expression, low levels promoting cell cycle progression but elevated levels cell cycle exit (*Radhakrishnan et al., 2004*; *Shats et al., 2017*; *Wang et al., 2007*). Our results are consistent with this observation. *E2f1* is expressed at a low level in the proliferative progenitors, and its expression peaks concomitant to cell cycle exit. Also, Ebf1 and Insm1 have earlier been associated with cell cycle exit and delamination of neuronal progenitors, as well as differentiation and migration of post-mitotic precursors (*Faedo et al., 2017*; *Garcia-Dominguez et al., 2003*; *Monaghan et al., 2017*; *Tavano et al., 2018*). Thus, the mechanisms that guide cell cycle exit and other cell biological events of neurogenesis may also contribute to activation of neuronal selector TF expression and adoption of the neuronal phenotype.

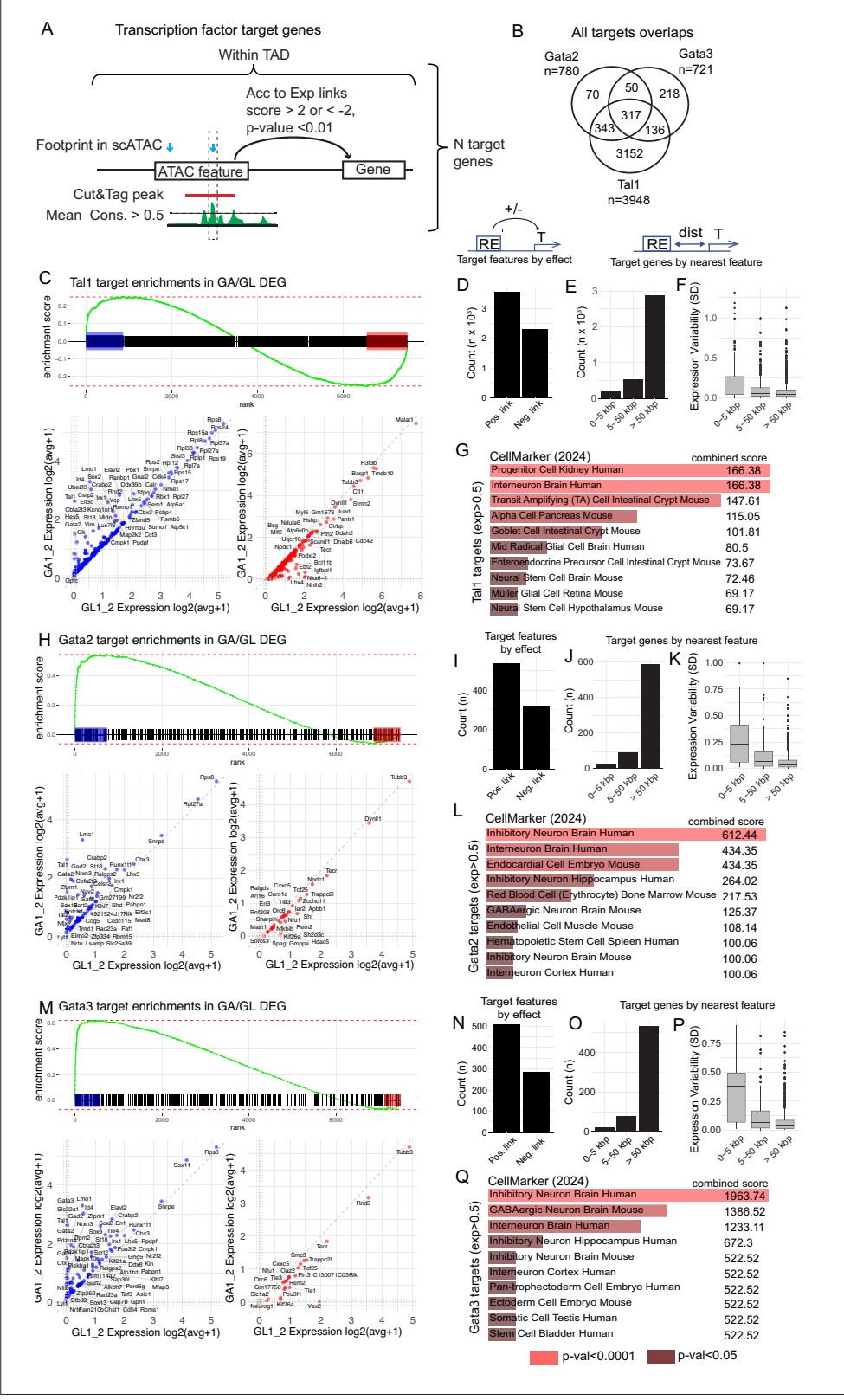

**Figure 6.** Genomic targets of the GABAergic selector transcription factors (TFs). (**A**) Schematic explaining the strategy of identifying the targets of Tal1, Gata2, and Gata3 selector TFs. Within the TAD containing a gene, features overlapping a CUT&Tag peak for the selector TF, and a footprint for the TF at a position with weighted mean conservation score >0.5 are identified. Genes linked to features fulfilling these conditions are considered

*Figure 6 continued on next page*

*Figure 6 continued*

target genes. For linkage, the Spearman correlation-based LinkPeaks score between the feature targeted by the selector TF and expression of the gene is required to be >2 (positive effect link) or <−2 (negative effect link) with a p-value <0.01. (**B**) Number of target genes and the overlap between Gata2, Gata3, and Tal1 target genes. (**C**) Gene set enrichment analysis (GSEA) of Tal1 targets. Mouse genes are ranked by the difference in the expression in GA1–2 vs GL1–2 cell groups (log2 avg FC). Tal1 target genes are indicated with black lines. In leading edges, the GA1–2-enriched target genes are highlighted in blue and GL1–2-enriched genes in red. Scatterplots show the expression of the target genes in both edges. (**D–F**) Characterization of Tal1 target genes. (**D**) Count of target features by the positive and negative effect link. (**E**) Count of target genes by the nearest linked feature, stratified by the nearest feature distance bins as indicated. (**F**) The variability of target gene expression in rV2 lineage cell clusters, stratified by the nearest feature distance bins. (**G**) Top terms in CellMarker gene set database using the list of Tal1 target genes with exp >0.5 (log1p) in GA1–2 or GL1–2 cell groups or both. (**H**) GSEA of Gata2 targets, as in (**C**). (**I–L**) Characterization of Gata2 target genes, as in (**D–G**). (**M**) GSEA of Gata3 targets, as in (**C**). (**N–Q**) Characterization of Gata3 target genes, as in (**D–G**).

## Selector TFs directly autoregulate themselves and cross-regulate each other

Positive feedback loops are commonly found in the gene regulatory networks involving developmental selector TFs. The positive feedback loops can ensure stable cell fate decisions. Tal and Gata TFs have been shown to autoregulate their own gene expression in haematopoietic cells (*Wilson et al., 2010*). Our results suggest that autoregulation is also a prominent feature of the control of *Tal1*, *Gata2*, and *Gata3* expression in the developing brain. This is also supported by loss-of-function studies in the mice, where Gata2 and Gata3 were shown to be dispensable for the initial induction of the *Tal1* expression, but required for its maintenance (*Lahti et al., 2016*). Reciprocally, Tal1 is required for the maintenance of *Gata2* and *Gata3* expression in the embryonic r1. At maintaining their expression by positive feedback, these TFs appear to work together, potentially as a physical complex involving *Lmo1* (*Ono et al., 1998*). In addition to the positive cross-regulation of Tal1, Gata2, and Gata3, these TFs also interact with the genes of glutamatergic lineage-specific TFs, such as *Vsx2* and *Lhx4*, expressed in the glutamatergic precursors. We envision negative feedback allowing cross-repression between the Tal1, Gata2, Gata3, Lmo1 network, and Vsx2 in the cells where both alternative fate selectors are expressed. Auto- and cross-regulation would ensure robust patterns of selector TF expression, also after the initial inputs for their activation have ceased.

TFs often have context-dependent activating or repressive effects on their targets, and our data on chromatin occupancy cannot distinguish between the two possible outcomes. However, according to our scRNA-seq and earlier studies in the developing spinal cord V2 domain (*Francius et al., 2016*; *Karunaratne et al., 2002*), the time of co-expression of *Gata2/3* and *Vsx1/2* is short. In the rV2, *Vsx1* is expressed in the rV2 common precursors, where *Tal1*, *Gata2*, and *Gata3* expression is still low. Parsimonious explanations would be the negative cross-regulation and differential regulation of Lmo1. Furthermore, Vsx1 has been shown to repress the expression of *Tal1* in the zebrafish spinal cord V2a interneurons (*Zhang*

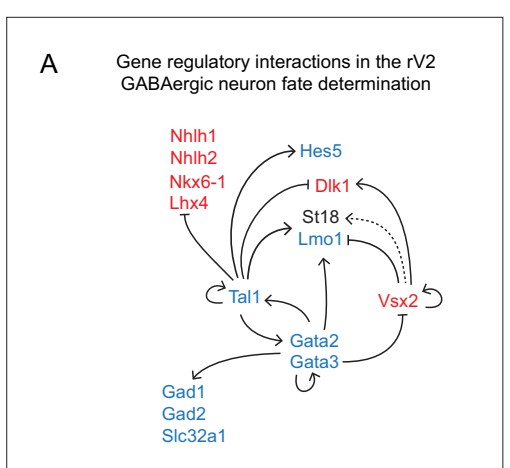

**Figure 7.** Proposed gene regulatory network guiding the GABA- vs glutamatergic fate selection in the rV2. The genes expressed in GABAergic cells (GA1–2) are in blue and the genes expressed in glutamatergic cells (GL1–2) in red. *St18* is expressed in both GA1–2 and GL1–2 cell groups, but its expression in GL1–2 is weak. Arrows represent positive regulation and were drawn when the regulator and its target mRNAs were co-expressed in the same cell group. Blunt arrows represent negative regulation and were drawn when the regulator and the target expression were not found in the same cell group.

The online version of this article includes the following figure supplement(s) for figure 7:

**Figure supplement 1.** Examples of definitions of selector gene target features and target genes.

*et al., 2020*). *Tal1* overexpression results in reduction of *Vsx2* expression in mouse ventral spinal cord V2a/V2b interneurons (*Muroyama et al., 2005*) and the overexpression of *Gata3* also results in the reduction of the number of *Vsx2* expressing cells in the chicken ventral spinal cord (*Karunaratne et al., 2002*).

### Targets of the selector TFs

In addition to establishing cell-type-specific patterns of selector TF expression, the selector TFs are thought to be responsible for realization of neuronal identities. Our results show that a combinatory function of the selector TFs is important for adopting a GABAergic identity. This is supported by similar fate transformations observed in *Tal1* and *Gata2/Gata3* mutant embryos (*Lahti et al., 2016*). Importantly, our results here also highlight differences between Tal1, Gata2, and Gata3 target genes. Most prominently, the activity of Tal1 is not as clearly biased towards genes expressed in the GABAergic branch as are the genes regulated by Gata2 and Gata3. In general, the Tal1, Gata2, and Gata3 selector TFs appear to regulate not only GABAergic subtype-specific genes, but also genes expressed more broadly. This suggests that the function of the fate selectors is not restricted to regulation of the neuronal subtype properties such as the neurotransmitter phenotype, but they also participate in enabling the overall cellular function.

### Concluding remarks

Our work provides insights into the complex gene regulatory networks regulating spatially and temporally controlled cell fate choices in the developing mouse brainstem. In addition to addressing principles of cell differentiation, our work may also contribute to understanding of neuropsychiatric traits and disorders. The anterior brainstem is an important centre for behavioural control, and many of the behavioural disorders have a developmental basis. We have shown that, in mice, a failure in the development of Tal1-dependent neurons results in severe ADHD-like behavioural changes (*Morello et al., 2020b*). It is therefore possible that variants in the genomic regions important for gene regulation during brainstem neuron differentiation can lead to altered brain development and predisposition to neuropsychiatric disorders.

### Caveats of the study

Partly due to technical limitations, some of the TF–chromatin interactions are inferred only based on scATAC-seq footprint. Some of the footprints may signify the binding sites of several different TFs, although mostly of the same TF family, and the exact interacting TF may not be identified. In addition, chromatin interaction is only suggestive of a regulatory function. Our interpretations of the biological processes related to the selector TF expression are based on the known functions of the regulatory TFs. However, many of these functions are unknown.

## Materials and methods

### scRNA-seq samples

scRNA-seq sample preparation is described in detail in *Morello et al., 2020a*. Outbred mouse strains were used in this study. The scRNA-seq samples were from E12.5 ICR mouse embryos. The GEO and BioProject IDs of the data are listed above.

### Processing of scRNA-seq data

scRNA-seq data processing per brain region is described in detail in (https://github.com/ComputationalNeurogenetics/rV2_bifurcation, copy archieved at *Kilpinen, 2025*) with main steps outlined below.

scRNA-seq data was read into *Seurat* objects from 10x *filtered_feature_bc* matrix by using Ensembl gene IDs as features. Biological replicates from E12.5 R1 scRNA-seq were merged as *Seurat* objects. After merging, quality control (QC) was performed by calculating *mt-percentage* and filtering based on *nFeature_RNA* and *percent.mt*. Data was then log-normalized, top 3000 most variable features

were detected, and finally, data was scaled and *mt-percentage*, *nFeature_RNA*, *nCount_RNA* were set to be regressed out.

Data was then run through PCA using 2000 variable features, and *Seurat RunUMAP* with selected top PCA dimensions was used to generate UMAP for further analysis. *Seurat* functions *FindNeighbors* and *FindClusters* were applied for neighbourhood and community detection-based clustering.

### scATAC-seq samples

Embryonic brains were isolated from E12.5 wild-type (NMRI) mouse embryos (*n* = 2), and ventral R1 pieces (E12.5) were separated. Nuclei were isolated using the 10X Genomics Assay for Transposase Accessible Chromatin Nuclei Extraction from fresh tissues protocol CG000212. Chromium Single Cell Next GEM ATAC Solution (10X Genomics, 1000162) was used for the capture of nuclei and the accessible chromatin capture. In chip loading, the nuclei yield was set at 5000 nuclei per sample. Chromium Single Cell Next GEM ATAC Reagent chemistry v1.1 was used for the DNA library preparation. Sample libraries were sequenced on Illumina NovaSeq 6000 system using read lengths: 50 bp (Read 1N), 8 bp (i7 Index), 16 bp (i5 Index), and 50 bp (Read 2N). The reads were quality filtered and mapped against the *Mus musculus* genome *mm10* using Cell Ranger ATAC v1.2.0 pipeline.

### Processing of scATAC-seq data

In this study, we defined feature space as described previously (*Kilpinen et al., 2023*), except that we used both E14.5 (*Kilpinen et al., 2023*) and E12.5 scATAC-seq samples to further increase resolution to detect features from rare cell population and cellular clade cut height was defined based on smallest clade so that none of the clades contained less than 100 cells (*h* = 8).

These jointly defined features were then used in the processing of the E12.5 rhombomere 1 scATAC-seq data. Data was subjected to QC, binarization and LSA, steps described in more detail in (https://github.com/ComputationalNeurogenetics/rV2_bifurcation, copy archieved at *Kilpinen, 2025*) largely following steps previously described (*Stuart et al., 2021*; *Stuart et al., 2019*). Similarly, scATAC-seq data was integrated with scRNA-seq using an approach already described (*Stuart et al., 2019*).

The UMAP projection based on scATAC-seq data was done with *Seurat* using SVD components 2–30 for cells with a reliability score from scRNA-seq integration >0.5. The first SVD component was omitted due to the association with read depth. Neighbourhood and community detection were made with Seurat functions *FindNeighbors*(dims = 2:30) and *FindClusters*(resolution = 1, algorithm = 4).

### Subsetting and subclustering the rV2 lineage

The rV2 lineage was subset from clustered E12.5 R1 scRNA-seq data (*Figure 1—figure supplement 1A*) as well as from E12.5 R1 scATAC-seq data (*Figure 1—figure supplement 2A*). Subsetting was based on the expression levels of known marker genes (*Ccnb1*, *Nkx6-1*, *Vsx2*, *Slc17a6*, *Tal1*, *Gad1*). scATAC-seq clusters 1, 6, 8, 12, 13, 19, 20, 24, and 26 were considered to consist of rV2 lineage. Clusters 13 and 19 were considered to be progenitors and were named PRO1 and PRO2, respectively. Cluster 26 was considered to consist of common precursors and was named CO. Clusters tentatively belonging to common precursors, the GABAergic (1, 6, 20, and 24), and the glutamatergic branches (8 and 12) were further subjected to Seurat *FindSubClusters* to reveal higher-resolution cell types. This resulted in the splitting of CO into clusters CO1 and CO2, the GABAergic branch into six clusters named GA1–GA6, and the glutamatergic branch into five clusters GL1–GL5.

### rV2 cluster correspondence between scATAC-seq and scRNA-seq

To evaluate the overlap of marker genes between scRNA-seq and scATAC-seq clusters (*Supplementary file 1*; *Supplementary file 2*), we performed a hypergeometric test and visualized the results using a heatmap. The scRNA-seq and scATAC-seq cluster markers were filtered with adj. p-value <0.05 and top 25 based on log2FC. An intersection of unique gene names was identified between scRNA-seq and scATAC-seq objects and its size, *N*, was used as the total gene pool for the hypergeometric test.

### VIA pseudotime analysis

For VIA (*Stassen et al., 2021*) based pseudotime analysis rV2-lineage embeddings coordinates from dimensions 2:30, as well as cluster identities as colours, were passed from Seurat object to

Python-based VIA. Gene expression levels were used to calculate the pseudotime value in both scRNA-seq and scATAC-seq. In the case of scATAC-seq, these were the interpolated values from integration with the scRNA-seq (*Stuart et al., 2019*). In the case of scATAC-seq, PRO1 was chosen as the root cell population, and for scRNA-seq, cluster 22 was chosen as root cell population.

## Sequence conservation

Genome conservation data was derived from UCSC database at https://hgdownload.cse.ucsc.edu/goldenPath/mm10/phastCons60way/, which contains compressed phastCons scores for multiple alignments of 59 vertebrate genomes to the mouse mm10 genome (*Siepel et al., 2005*). Nucleotide was considered conserved if phast-score was at least 0.5 (range 0–1). The scATAC-seq features and other selected regions of study were examined for conservation level in comparison to the whole genome, using Fisher's exact test. Conservation was also calculated for exon and intron regions within all chromosomes to demonstrate the accuracy of this method. For sequence conservation at the TFBS-motif locations, positional weight matrix (PWM) of the motif was used to define weight of each nucleotide based on max probability at that nucleotide location in PWM and that weight was used in calculation of the weighted mean of conservation score over the nucleotides at TFBS locations.

## TAD

TADs of the chromatin, defined by *tadmap* by aggregating HiC datasets (*Singh and Berger, 2021*), were used to limit the search for cCREs.

## Feature accessibility over pseudotime

Features within TAD of the selector gene were considered to be cCRE if they had linkage correlation with expression of the gene with abs(zscore) >2 and p-value <0.05. Accessibility changes of cCREs were visualized as spline-smoothed heatmaps over the pseudotime. Spline smoothing was calculated over entire rV2 cell groups in order of PRO1–2, CO1–2, GA1–2, GA3–4, GA5–6, GL1–2, GL3–4, and GL5.

## TF-footprinting from scATAC-seq reads

TF-footprinting of scATAC-seq data was performed by using the TOBIAS framework (*Bentsen et al., 2020*) with Snakemake pipeline. As a motif collection, we used Hocomoco version 12 (*Vorontsov et al., 2024*). The motif for INSM1 was added to the Hocomoco v12 collection from the Hocomoco v11 collection as it was seen relevant in earlier research. The suitability of mixing motifs from the two versions of the Hocomoco was checked in correspondence with the Hocomoco team.

For the purpose of TF-footprinting, we combined cell groups as follows to increase signal-to-noise ratio: PRO1 and PRO2 (PRO1–2), CO1 and CO2 (CO1–2), GA1 and GA2 (GA1–2), GA3 and GA4 (GA3–4), GA5 and GA6 (GA5–6), GL1 and GL2 (GL1–2), and GL3 and GL4 (GL3–4). GL5 was kept as is. E12R1 BAM files were merged, and PCR duplicates removed with Picard (https://broadinstitute.github.io/picard/). A BAM file was then split into BAM file per combined cell group with Sinto (*Stuart, 2024*; https://timoast.github.io/sinto/).

TOBIAS was run with parameters given in https://github.com/ComputationalNeurogenetics/rV2_bifurcation (copy archieved at *Kilpinen, 2025*) and the cell group and genomic location specific footprint results were imported into an SQLite database (total of 49,390,084 records).

Footprinting data was visualized through R ggplot based dotplots per selected features, with filtering for TFBS-motifs having weighted average conservation value >0.5, accessibility of the feature >0.06 and expression of the TF in question >1.2 (log1p) in any of the PRO1–2, CO1–2, GA1–2, or GL1–2 groups.

## RNAscope mRNA in situ hybridization

E12.5 wild-type (NMRI) mouse embryos were dissected and fixed in 4% paraformaldehyde (Sigma-Aldrich, Cat#P6148) in 1× PBS for 36 hr at room temperature. The fixed brains were embedded in paraffin (Histosec Pastilles (without DMSO, Cat#115161)), and sectioned on a Leica microtome (Leica RM2255) at 5 µm. RNAscope Multiplex Fluorescent Reagent Kit v2 (ACD, Cat#323270) was used to detect RNA transcripts. The paraffin sections were deparaffinized in xylene and ethanol series, treated with RNAscope Hydrogen Peroxide (ACD, Cat#322381) for 10 min, and subjected to antigen

retrieval in 1× Target Retrieval Solution (ACD, Cat#322000) for 15 min at 99–102°C. After rinsing and dehydration, hydrophobic barriers were drawn using an ImmEdge Pen (ACD, Cat#310018), and slides were dried for 30 min or overnight. Protease Plus (ACD, Cat#322381) was applied for 30 min at 40°C, followed by hybridization of probes at 40°C for 2 hr. Probes included mouse *Ebf1* (ACD, Cat#433411, C2), *E2f1* (ACD, Cat#431971, C1), *Insm1* (ACD, Cat#43062, C1), *Sox4* (ACD, Cat#471381, C2), *Tal1* (ACD, Cat#428221, C4), and *Tead2* (ACD, Cat#42028, C3). Positive (ACD, Cat#320881) and negative (ACD, Cat#320871) control probes were included. C1 probes were used undiluted; C2, C3, and C4 probes were diluted 1:50 in C1 probe or Probe Diluent (ACD, Cat#300041). Slides were washed in Wash Buffer (ACD, Cat#310091) and stored in 5× saline-sodium citrate (SSC) buffer overnight. Amplification was conducted using AMP1, AMP2, and AMP3 reagents (30, 30, and 15 min) at 40°C. Signal detection was performed with HRP channels (C1, C2, and C3) and TSA Vivid Fluorophores: TSA 520 (ACD, Cat#323271),TSA 570 (ACD, Cat#323272), and TSA 650 (ACD, Cat#323273), diluted 1:1500 in RNAscope Multiplex TSA Buffer (ACD, Cat#322809). Each HRP step included blocking and washing. Slides were counterstained with DAPI (30 s), mounted with ProLong Gold Antifade Reagent (Thermo Fisher, Cat#P36934), and imaged using a Zeiss AxioImager 2 microscope with Apotome 2 structured illumination. Images were processed in ImageJ, and figures prepared in Adobe Illustrator CC 2020. Slides were stored at 4°C and imaged within 2 weeks.

## Image segmentation and downstream analysis

We trained a Stardist (*Schmidt et al., 2018*; *Weigert et al., 2020*) model to segment the images. For this purpose, we manually painted cell nucleus onto 10 representative microscopy images of E12.5 rV2 area based on DAPI channel. The trained model was then used to segment the analysis images and to return Regions of interestROIs as ImageJ ROI coordinates. Regions of interest (ROIs) were imported into R for analysis. Fluorescence signals of the three channels in each image were quantified separately. The VS was defined manually using three points and calculating circle boundary passing these points, and the distances of individual cells from the VS were calculated in units of average cell diameter (avg. across all images). Fluorescence signals were filtered to remove background noise using a quantile threshold of 0.25. Raw fluorescence intensity was measured as the number of fluorescence dots per cell. To analyse patterns across biological replicates, fluorescence data from three independent experiments were included in the same aggregate analysis. Fluorescence intensities were normalized across biological replicates for each channel to account for variations in signal strength and aligned to ensure uniform spatial scaling. Aggregated data were visualized by calculating a smoothed sliding mean (window size = 12) fluorescence trend for each channel.

## CUT&Tag

Embryonic brains were isolated from E12.5 wild-type (ICR) mouse embryos, and ventral R1 pieces (E12.5) were separated. The ventral R1 pieces from the embryos of the same litter were pooled and dissociated. The cells were then divided between the samples. Embryos from different E12.5 mouse litters were used as biological replicates in CUT&Tag. The samples were processed following the CUT&Tag-It Assay Kit protocol (Active Motif, cat # 53170). The following primary antibodies were used: rabbit anti-Gata2 (Abcam, ab109241), rabbit anti-Gata3 (Boster, M00593), rabbit anti-Tal1 (Abcam, ab75739), rabbit anti-Vsx2 (Proteintech, 25825-1-AP-20), rabbit anti-Ebf1 (*Boller et al., 2016*), rabbit anti-Insm1 (Nordic BioSite, ASJ-IO4DE3-50), rabbit anti-Tead2 (Biorbyt, orb382464), rabbit anti-IgG (Cell Signalling, 66362), and rabbit anti-H3K4me3 (Cell Signalling, 9751). The anti-Ebf1, anti-Tal1, anti-IgG, and anti-H3K4me3 antibodies were tested on Cut-and-Run or ChIP-seq previously (*Boller et al., 2016*; *Courtial et al., 2012*) and Cell Signalling product information. The anti-Gata2 and anti-Gata3 antibodies are ChIP-validated [(*Ahluwalia et al., 2020*) and Abcam product information]. There are no previous results on ChIP, ChIP-seq, or CUT&Tag with the anti-Insm1, anti-Tead2, and anti-Vsx2 antibodies used here. The specificity and nuclear localization have been demonstrated in immunohistochemistry with anti-Vsx2 (*Ahluwalia et al., 2020*) and anti-Tead2 (Biorbyt product information). We observed good correlation between replicates with anti-Insm1, similar to all antibodies used here, but its specificity to target was not specifically tested.

All primary antibodies were used at 1:80 dilution. Sample and sequencing library preparation was done following the recommended protocols of the assay. The DNA libraries were multiplexed at 10 nM equimolar ratios and sequenced on NextSeq 500 or Aviti. 75 bp paired-end sequences

were obtained. The demultiplexed reads were quality filtered and aligned against the *Mus musculus* genome *mm10*, using the nf-core (*Ewels et al., 2020*) pipeline nf-core/cutandrun.

## Comparison of TF binding detected by CUT&Tag and footprinting

To quantify how CUT&Tag and footprinting corroborate TF binding, we performed an analysis similar to the established one (*Eastman et al., 2025*). Per each TF with CUT&Tag data, we calculated (1) total number of CUT&Tag consensus peaks; (2) total number of bound TFBS; (3) percentage of CUT&Tag consensus peaks overlapping bound TFBS; and (4) percentage of bound TFBS overlapping CUT&Tag consensus peak (*Supplementary file 6*). TFBS in question was considered 'bound' if it had a positive footprint in at least one of the studied cell groups (PRO1_2, CO1_2, GA1_2, GA3_4, GA5_6, GL1_2, GL3_4, GL5). Overlaps were defined with min = 1.

We also performed an analysis of the footprint locations of a TF within the CUT&Tag peaks of the TF in question. For each TF all, the CUT&Tag consensus peaks with bound TFBS in at least one of the studied cell groups (PRO1_2, CO1_2, GA1_2, GA3_4, GA5_6, GL1_2, GL3_4, GL5) were found. The peak lengths were normalized (to 0–1 from peak start–end) and the density of overlapping bound footprint positions plotted over the normalized length of the peaks (*Figure 3—figure supplement 6*).

## Analysis of the TFs interacting with the selector gene cCREs

To search for regulators of selector genes, we first searched for TFs per each selector gene per cell group with conditions as follows:

1. The TF has a footprint in any of the cCREs linked to the selector gene with LinkPeaks abs($z$-score) >2 and p-value <0.05, and
2. The weighted mean of the conservation of the motif location is >0.5.
3. The TF is expressed >1.2 (log1p) on average in the cell group in question.

Defining the common regulators between the selector genes was done by comparison of the regulators found for individual selectors based on their intersects. Statistical significance levels reported for individual regulators are defined as minimum p-value for cCRE to gene linkage for each regulator per each selector gene and maximum p-value over all selectors in the intersect. To assess whether the observed overlap in regulators among the three selector genes was greater than expected by chance, we performed a permutation-based test. First, we determined the actual set of regulators for each of the three selector genes and calculated the size of their intersection. Next, to construct a null model, we repeatedly and randomly selected three sets of regulators (of the same sizes as the original sets) from the complete universe of candidate regulators, recalculating the intersection size each time. With 1,000,000 iterations, we obtained a distribution of intersection sizes expected under random selection. The empirical p-value was then estimated as the proportion of these randomized trials in which the intersection size was equal to or larger than the observed intersection. If no randomized trial matched or exceeded the observed intersection size, we reported the p-value as less than an upper bound (e.g.,<1/(N+1)), reflecting the rarity of the observed overlap under the assumed null hypothesis.

## Selector gene target analysis

To uncover general functional roles of the selector genes at the bifurcation point, we conducted a GSEA (*Mootha et al., 2003*) of the putative target genes of Tal1, Gata2, Gata3, and Vsx2 over the axis of avg_log2FC of mouse genes in between GA1–2 and GL1–2. For the GSEA axis, only genes having an adjusted p-value <0.01 in the DEG test (Seurat FindMarkers with Wilcoxon) and an expression proportion of at least 0.1 in either group were taken into account.

Selector TF target genes were defined according to the following conditions:

1. The selector gene (TF) motif is associated with a footprint in either GA1–2 or GL1–2.
2. The motif situates in a scATAC-seq feature which is linked to the target gene with LinkPeaks abs($z$-score) >2 and p-value <0.01.
3. The weighted mean of the conservation of the motif location is >0.5.
4. There is a CUT&Tag peak of the selector gene overlapping the feature.

As our GSEA axis should be considered relevant from both ends, one end signifying targets over-expressed in GA1–2 and another in GL1–2, we additionally looked for the leading edge genes from

both ends (traditionally called leading and trailing edges in GSEA analysis). We defined GA1–2 leading edge genes to be the ones before the GSEA enrichment score gains its maximum, and GL1–2 leading edge genes to be ones after the GSEA enrichment score gains its minimum.

## Comparison of selector gene targets and cell type markers

The selector gene target list was subset, selecting the genes with exp_avg_GL1–2 >0.5 (log1p) OR exp_avg_GA1–2 >0.5 (log1p). The resulting gene list was used as input in the online Enrichr tool (*Xie et al., 2021*). The results against the CellMarker database (*Hu et al., 2023*) were extracted.

## Cell-type-specific target genes of selectors

Cell-type-specific genes were defined as log2FC >0.5; exp_avg_GL1–2 <0.5 (log1p); exp_avg_GA1–2 >0.5 (log1p) for GABAergic neuron-specific genes and log2FC <−0.5; exp_avg_GL1–2 >0.5 (log1p); exp_avg_GA1–2 <0.5 (log1p) for glutamatergic neuron-specific genes. The Tal1, Gata2, and Gata3 target gene lists were subset using these criteria.

## ChromVAR analysis with Hocomoco motifs

ChromVAR analysis (*Schep et al., 2017*) was performed using Hocomoco (*Vorontsov et al., 2024*) v12 motifs and an INSM1 motif lifted from Hocomoco v11 by using functions built into Signac package. The motifs whose chromatin accessibility correlated with the RNA expression of their respective genes (Spearman correlation >0.5 and p-value <0.01) were identified, and their ChromVAR *z*-scores across rV2 lineage cell groups were aggregated by mean to a level of TF and then visualized using Complex-Heatmap R package applying clustering method 'complete' and the clustering distance as 'euclidean'.

## Acknowledgements

We thank Outi Kostia for expert technical assistance and Iiris Chrons for help with the bioinformatic analyses. We thank Arto Viitanen for his guidance with methods of image segmentation. We acknowledge the services of the FIMM Single Cell Analysis Unit, University of Helsinki. This work was supported by grants from the Academy of Finland, the Sigrid Juselius Foundation, and the Magnus Ehrnrooth Foundation.

## Additional information

### Funding

| Funder | Grant reference number | Author |
| --- | --- | --- |
| Research Council of Finland | 331261/362600 | Sami Kilpinen<br>Lassi Virtanen<br>Silvana Bodington Celma<br>Amos Bonsdorff<br>Heidi Heliölä<br>Kaia Achim<br>Juha Partanen |
| Sigrid Jusélius Foundation | 1119 | Sami Kilpinen<br>Lassi Virtanen<br>Silvana Bodington Celma<br>Amos Bonsdorff<br>Heidi Heliölä<br>Kaia Achim<br>Juha Partanen |
| Magnus Ehrnroothin Säätiö | 2021/2024 | Sami Kilpinen<br>Lassi Virtanen<br>Silvana Bodington Celma<br>Amos Bonsdorff<br>Heidi Heliölä<br>Kaia Achim<br>Juha Partanen |

| Funder | Grant reference number | Author |
|---|---|---|

The funders had no role in study design, data collection, and interpretation, or the decision to submit the work for publication.

## Author contributions

Sami Kilpinen, Conceptualization, Data curation, Formal analysis, Supervision, Visualization, Methodology, Writing – original draft, Writing – review and editing; Lassi Virtanen, Software, Formal analysis, Visualization; Silvana Bodington Celma, Data curation, Investigation, Methodology, Writing – original draft; Amos Bonsdorff, Heidi Heliölä, Software, Formal analysis; Kaia Achim, Conceptualization, Supervision, Validation, Methodology, Writing – original draft, Writing – review and editing; Juha Partanen, Conceptualization, Resources, Supervision, Funding acquisition, Validation, Visualization, Project administration, Writing – review and editing

## Author ORCIDs

Sami Kilpinen ⬚ https://orcid.org/0000-0003-3444-2539
Kaia Achim ⬚ https://orcid.org/0000-0003-3723-4065
Juha Partanen ⬚ https://orcid.org/0000-0001-8850-4825

## Ethics

This research involves materials collected from laboratory mice, following the ethical research guidelines given by EU in Directive 2010/63/EU on the protection of animals used for scientific purposes. The study was approved by the Laboratory Animal Center, University of Helsinki (permit number KEK23-032). Mice were handled according to the procedures approved by the Laboratory Animal Center, University of Helsinki. The 3R principles (reduction, replacement, refinement) were followed in study design.

Reviewer #1 (Public review): https://doi.org/10.7554/eLife.105867.3.sa1
Reviewer #2 (Public review): https://doi.org/10.7554/eLife.105867.3.sa2
Author response https://doi.org/10.7554/eLife.105867.3.sa3

# Additional files

## Supplementary files

Supplementary file 1. scRNA-seq cluster markers.

Supplementary file 2. scATAC-seq cluster markers.

Supplementary file 3. Features and cCREs per selector gene.

Supplementary file 4. Cross-comparison to a priori known literature.

Supplementary file 5. Conservation analysis results.

Supplementary file 6. Footprint and CUT&Tag overlap.

Supplementary file 7. Selector gene target analysis results.

Supplementary file 8. Cell type specificity of selector TF target genes and the analysis of selector genes' common targets.

MDAR checklist

## Data availability

The E12.5 mouse rhombomere 1 scRNA-seq and E12.5 scATAC-seq reads generated in our study are deposited in SRA. The BioProject IDs of the E12.5 R1 scRNA-seq data are GEO GSE157963; BioProject PRJNA663556 E12.5 R1 samples SAMN16134208, SAMN16134206, SAMN16134205, and SAMN16134207; and E12.5 R1 scATAC-seq: BioProject PRJNA929317, samples SAMN32957134 and SAMN32957135. The E12.5 r1 CUT&Tag data is available in GEO under accession ID GSE298231. All original code with used parameter values will be deposited in GitHub at https://github.com/ComputationalNeurogenetics/rV2_bifurcation (copy archieved at *Kilpinen, 2025*). Any additional information required to reanalyse the data reported in this paper is available upon request. The exact cell number, read yields, and read quality for each sample are found in https://github.com/Computationa

lNeurogenetics/rV2_bifurcation/tree/main/seq_quality_reports. Batch effect was evaluated not to be meaningful based on the distribution of cells from biological replicates in the UMAPs of both r1 E12.5 scRNA-seq and scATAC-seq.

The following dataset was generated:

| Author(s) | Year | Dataset title | Dataset URL | Database and Identifier |
|-----------|------|---------------|-------------|-------------------------|
| Kilpinen S, Achim K, Partanen J | 2025 | Gene regulatory mechanisms guiding bifurcation of inhibitory and excitatory neuron lineages in the anterior brainstem | https://www.ncbi.nlm.nih.gov/geo/query/acc.cgi?acc=GSE298231 | NCBI Gene Expression Omnibus, GSE298231 |

The following previously published datasets were used:

| Author(s) | Year | Dataset title | Dataset URL | Database and Identifier |
|-----------|------|---------------|-------------|-------------------------|
| Kilpinen S, Achim K | 2023 | Chromatin accessibility and RNA expression in E14.5 mouse neurons | https://www.ncbi.nlm.nih.gov/bioproject/PRJNA929317 | NCBI BioProject, PRJNA929317 |
| Partanen J, Achim K, Borshagovski D, Samir-Sadik O | 2020 | Single-cell RNAsequencing of the ventral rhombomere 1 of the developing mouse | https://www.ncbi.nlm.nih.gov/geo/query/acc.cgi?acc=GSE157963 | NCBI Gene Expression Omnibus, GSE157963 |
| Morello et al. | 2020 | Single-cell RNAsequencing of the ventral rhombomere 1 of the developing mouse [10X Genomics] (house mouse) | https://www.ncbi.nlm.nih.gov/bioproject/PRJNA663556 | NCBI BioProject, PRJNA663556 |

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
