## [Editor Report · eLife Assessment]

This work is a **important** resource for hypothesis testing of candidate upstream transcriptional regulatory factors that control the spatiotemporal expression of selector genes and their targets for GABAergic vs glutamatergic neuron fate in the anterior brainstem. Extensive high-quality datasets were generated and state of the art computational methods were **convincingly** implemented to identify candidate regulatory elements. The work will be of interest to biologists working to understand neuronal gene regulatory networks.

---

## [Referee Report · Reviewer #1 (Public review)]

The objectives of this research are to understand how key selector transcription factors, Tal1, Gata2, Gata3, determine GABAergic vs glutamatergic neuron fate from the rhombencephalic V2 precursor domain and how their spatiotemporal expression is controlled by upstream regulators. Toward these goals, the authors have generated an impressive array of scRNA, scATAC-seq, and CUT&Tag datasets obtained from dissociated E12.5 ventral R1 dissections. The rV2 was subsetted with well-known markers. The authors use an extensive set of computational approaches to identify temporal patterns of chromatin accessibility, TF motif binding activities (footprints), gene expression and regulatory motifs at the different selector gene loci. These analyses are used to predict upstream regulators, candidate accessible CREs, and DNA binding motifs through which the selectors may be controlled in rV2 by upstream regulators. Further analyses predict auto- and cross-regulatory interactions for maintenance of selector expression and the downstream effectors of alternative transmitter identities controlled by the selectors. The authors have achieved their aim of making predictions about upstream and downstream selector TF regulatory networks; their conclusions and predictions are largely well supported. The work clearly illustrates the daunting gene regulatory complexity likely at play in controlling rV2 transmitter fate.

This is data-rich study and a valuable resource for future hypothesis testing, through perturbation approaches, of the many putative regulators and motifs identified in the study. The strengths of this work are the overall high quality of the datasets and in depth analyses. Through its comprehensive data and predictions, it is likely to have impact in advancing the understanding of GABAergic vs glutamatergic neuron fate decisions. The authors present a "simplified" gene regulatory model. However, the model does not illustrate the complexity of potential stage-specific upstream TF interactions with Tal1 and Vsx2 selector genes uncovered in TF footprinting analyses. While this seems nearly impossible to achieve given the plethora of potential functional TF inputs, the authors should consider assembling a focussed model by selectively illustrating the most robust, evidence-backed upstream TF input predictions, which are considered the strongest candidates for future hypothesis-driven perturbation experiments. It seems Insm1, Sox4, E2f1, Ebf1 and Tead2 TFs might be the strongest upstream candidates for future testing of Tal1 activation given the extensive analyses of their spatiotemporal expression patterns relative to Tal1, presented in Fig 4.

---

## [Referee Report · Reviewer #2 (Public review)]

Summary:

In this study, the authors seek to discover putative gene regulatory interactions underlying the lineage bifurcation process of neural progenitor cells in the embryonic mouse anterior brainstem into GABAergic and glutamatergic neuronal subtypes. The authors analyze single-cell RNA-seq and single-cell ATAC-seq datasets derived from the ventral rhombomere 1 of embryonic mouse brainstems to annotate cell types and make predictions or where TFs bind upstream and downstream of the effector TFs using computational methods. They add data on the genomic distributions of some of the key transcription factors, and layer these onto the single cell data to develop a model of the transcription factors interactions that define this fate choice.

Strengths:

The authors use a well-defined fate decision point from brainstem progenitors that can make two very different kinds of neurons. They already know the key TFs for selecting the neuronal type from genetic studies, so they focus their gene regulatory analysis on the mechanisms that are immediately upstream and downstream of these key factors. The authors use a combination of single-cell and bulk sequencing data, prediction and validation, and computation.

Weaknesses:

The study does not go as far as to experimentally test the transcription factor network from their model.

---

## [Author Response]

The following is the authors’ response to the original reviews

**Public Reviews:**

**Reviewer #1 (Public review):**
Summary:The objective of this research is to understand how the expression of key selector transcription factors, Tal1, Gata2, Gata3, involved in GABAergic vs glutamatergic neuron fate from a single anterior hindbrain progenitor domain is transcriptionally controlled. With suitable scRNAseq, scATAC-seq, CUT&TAG, and footprinting datasets, the authors use an extensive set of computational approaches to identify putative regulatory elements and upstream transcription factors that may control selector TF expression. This data-rich study will be a valuable resource for future hypothesis testing, through perturbation approaches, of the many putative regulators identified in the study. The data are displayed in some of the main and supplemental figures in a way that makes it difficult to appreciate and understand the authors' presentation and interpretation of the data in the Results narrative. Primary images used for studying the timing and coexpression of putative upstream regulators, Insm1, E2f1, Ebf1, and Tead2 with Tal1 are difficult to interpret and do not convincingly support the authors' conclusions. There appears to be little overlap in the fluorescent labeling, and it is not clear whether the signals are located in the cell soma nucleus.Strengths:The main strength is that it is a data-rich compilation of putative upstream regulators of selector TFs that control GABAergic vs glutamatergic neuron fates in the brainstem. This resource now enables future perturbation-based hypothesis testing of the gene regulatory networks that help to build brain circuitry.

We thank Reviewer #1 for the thoughtful assessment and recognition of the extensive datasets and computational approaches employed in our study. We appreciate the acknowledgment that our efforts in compiling data-rich resources for identifying putative regulators of key selector transcription factors (TFs)—*Tal1*, *Gata2*, and *Gata3*—are valuable for future hypothesis-driven research.

Weaknesses:Some of the findings could be better displayed and discussed.

We acknowledge the concerns raised regarding the clarity and interpretability of certain figures, particularly those related to expression analyses of candidate upstream regulators such as Insm1, E2f1, Ebf1, and Tead2 in relation to *Tal1*. We agree that clearer visualization and improved annotation of fluorescence signals are crucial to accurately support our conclusions. In our revised manuscript, we will enhance image clarity and clearly indicate sites of co-expression for Tal1 and its putative regulators, ensuring the results are more readily interpretable. Additionally, we will expand explanatory narratives within the figure legends to better align the figures with the results section.

**Reviewer #2 (Public review):**
Summary:In the manuscript, the authors seek to discover putative gene regulatory interactions underlying the lineage bifurcation process of neural progenitor cells in the embryonic mouse anterior brainstem into GABAergic and glutamatergic neuronal subtypes. The authors analyze single-cell RNA-seq and single-cell ATAC-seq datasets derived from the ventral rhombomere 1 of embryonic mouse brainstems to annotate cell types and make predictions or where TFs bind upstream and downstream of the effector TFs using computational methods. They add data on the genomic distributions of some of the key transcription factors and layer these onto the single-cell data to get a sense of the transcriptional dynamics.Strengths:The authors use a well-defined fate decision point from brainstem progenitors that can make two very different kinds of neurons. They already know the key TFs for selecting the neuronal type from genetic studies, so they focus their gene regulatory analysis squarely on the mechanisms that are immediately upstream and downstream of these key factors. The authors use a combination of single-cell and bulk sequencing data, prediction and validation, and computation.

We also appreciate the thoughtful comments from Reviewer #2, highlighting the strengths of our approach in elucidating gene regulatory interactions that govern neuronal fate decisions in the embryonic mouse brainstem. We are pleased that our focus on a critical cell-fate decision point and the integration of diverse data modalities, combined with computational analyses, has been recognized as a key strength.

Weaknesses:The study generates a lot of data about transcription factor binding sites, both predicted and validated, but the data are substantially descriptive. It remains challenging to understand how the integration of all these different TFs works together to switch terminal programs on and off.

Reviewer #2 correctly points out that while our study provides extensive data on predicted and validated transcription factor binding sites, clearly illustrating how these factors collectively interact to regulate terminal neuronal differentiation programs remains challenging. We acknowledge the inherently descriptive nature of the current interpretation of our combined datasets.

In our revision, we will clarify how the different data types support and corroborate one another, highlighting what we consider the most reliable observations of TF activity. Additionally, we will revise the discussion to address the challenges associated with interpreting the highly complex networks of interactions within the gene regulatory landscape.

We sincerely thank both reviewers for their constructive feedback, which we believe will significantly enhance the quality and accessibility of our manuscript.

**Recommendations for the authors:**

**Reviewer #1 (Recommendations for the authors):**
(1) The results in Figure 3 and several associated supplements are mainly a description/inventory of putative CREs some of which are backed to some extent by previous transgenic studies. But given the way the authors chose to display the transgenic data in the Supplements, it is difficult to fully appreciate how well the transgenic data provide functional support. Take, for example, the Tal +40kb feature that maps to a midbrain enhancer: where exactly does +40kb map to the enhancer region? Is Tal +40kb really about 1kb long? The legend in Supplemental Figure 6 makes it difficult to interpret the bar charts; what is the meaning of: features not linked to gene -Enh? Some of the authors' claims are not readily evident or are inscrutable. For example, Tal locus features accessible in all cell groups are not evident (Fig 2A,B). Other cCREs are said to closely correlate with selector expression for example, Tal +.7kb and +40kb. However, inspection of the data seems to indicate that the two cCREs have very different dynamics and only +40kb seems to correlate with the expression track above it. Some features are described redundantly such as the Gata2 +22 kb, +25.3 kb, and +32.8 kb cCREs above and below the Gata3 cCRE. What is meant by: The feature is accessible at 3' position early, and gains accessibility at 5' positions ... Detailed feature analysis later indicated the binding of Nkx6-1 and Ascl1 that are expressed in the rV2 neuronal progenitors, at 3' positions, and binding of Insm1 and Tal1 TFs that are activated in early precursors, at 5' positions (Figure 3C).

To allow easier assessment of the overlap of the features described in this study in reference to the transgenic studies, we have added further information about the scATAC features, cCREs and previously published enhancers, as well as visual schematics of the feature-enhancer overlaps in the Supplementary table 4. The Supplementary Table 4 column contents are also now explained in detail in the table legend (under the table). We hope those changes make the feature descriptions clearer. To answer the reviewer's question about the Tal1+40kb enhancer, the length of the published enhancer element is 685 bp and the overlapping scATAC feature length is 2067 bp (Supplementary Table 3, sheet *Tal1*, row 103).

The legend and the chart labelling in the Supplementary Figure 5 (formerly Supplementary figure 6) have been elaborated, and the shown categories explained more clearly.

Regarding the features at the Tal1 locus, the text has been revised and the references to the features accessible in all cell groups were removed. These features showed differences in the intensity of signal but were accessible in all cell groups. As the accessibility of these features does not correlate with *Tal1* expression, they are of less interest in the context of this paper.

The gain in accessibility of the +0.7kb and +40 kb features correlates with the onset of Tal1 RNA expression. This is now more clearly stated in the text, as " For example, the gain in the accessibility of Tal1 cCREs at +0.7 and +40 kb correlated temporally with the expression of Tal1 mRNA (Figure 2B), strongly increasing in the earliest GABAergic precursors (GA1) and maintained at a lower level in the more mature GABAergic precursor groups (GA2-GA6), " (Results, page 4). The reviewer is right that the later dynamics of the +0.7 and +40 cCREs differ and this is now stated more clearly in the text (Results, page 5, last chapter).

The repetition in the description of the *Gata2* +22 kb, +25.3 kb, and +32.8 kb cCREs has been removed.

The Tal1 +23 kb cCRE showed within-feature differences in accessibility signal. This is explained in the text on page 5, referring to the relevant figure 2A, showing the accessibility or scATAC signal in cell groups and the features labelled below, and 3C, showing the location of the Nkx6-1 and Ascl1 binding sites in this feature: "The Tal1 +23 kb cCRE contained two scATAC-seq peaks, having temporally different patterns of accessibility. The feature is accessible at 3' position early, and gains accessibility at 5' positions concomitant with GABAergic differentiation (Figure 2A, accessibility). Detailed feature analysis later indicated that the 3' end of this feature contains binding sites of Nkx6-1 and Ascl1 that are expressed in the rV2 neuronal progenitors, while the 5' end contains TF binding sites of Insm1 and Tal1 TFs that are activated in early precursors (described below, see Figure 3C)."

(2) Supplementary Figure 3 is not presented in the Results.

Essential parts of previous Supplementary Figure 3 have been incorporated into the Figure 4 and the previous Supplementary Figure omitted.

(3) The significance of Figure 3 and the many related supplements is difficult to understand. A large number of footprints with wide-ranging scores, many very weak or unbound, are displayed in the various temporal cell groups in different epigenomic regions of Tal1 and Vsx2. The footprints for GA1 and Ga2 are combined despite Tal1 showing stronger expression in GA1 and stronger accessibility (Figure 2). Many possibilities are outlined in the Results for how the many different kinds of motifs in the cCREs might bind particular TFs to control downstream TF expression, but no experiments are performed to test any of the possibilities. How well do the TOBIAS footprints align with C&T peaks? How was C&T used to validate footprints? Are Gata2, 3, and Vsx2 known to control Tal1 expression from perturbation experiments?

Figure 3 and related supplements present examples of the primary data and summarise the results of comprehensive analysis. The methods of identifying the selector TF regulatory features and the regulators are described in the Methods (Materials and Methods page 16). Briefly, the correlation between feature accessibility and selector TF RNA expression (assessed by the LinkPeaks score and p-value) were used to select features shown in the Figure 3.

We are aware of differences in *Tal1* expression and accessibility between GA1 and GA2. However, number of cells in GA2 was not high enough for reliable footprint calculations and therefore we opted for combining related groups throughout the rV2 lineage for footprinting.

As suggested, CUT&Tag could be used to validate the footprinting results with some restrictions. In the revised manuscript, we included analysis of CUT&Tag peak location and footprints similarly to an earlier study (Eastman et al. 2025). In summary, we analysed whether CUT&Tag peaks overlap locations in which footprinting was also recognized and vice versa. Per each TF with CUT&Tag data we calculated (a) Total number of CUT&Tag consensus peaks (b) Total number of bound TFBS (footprints) (c) Percentage of CUT&Tag overlapping bound TFBS (d) Percentage of bound TFBS overlapping CUT&Tag. These results are shown in Supplementary Table 6 and in Supplementary figure 11 with analysis described in Methods (Materials and Methods, page 19). There is considerable overlap between CUT&Tag peaks and bound footprints, comparable to one shown in Eastman et al. 2025. However, these two methods are not assumed to be completely matching for several reasons: binding by related/redundant TFs, antigen masking in the TF complex, chromatin association without DNA binding, etc. In addition, some CUT&Tag peaks with unbound footprints could arise from non-rV2 cells that were part of the bulk CUT&Tag analysis but not of the scATAC footprint analysis.

The evidence for cross-regulation of selector genes and the regulation of *Tal1* by *Gata2*, *Gata3* and *Vsx2* is now discussed (Discussion, chapter Selector TFs directly autoregulate themselves and cross-regulate each other, page 12-13). The regulation of *Tal1* expression by Vsx2 has, to our knowledge, not been earlier studied.

(4) Figure 4 findings are problematic as the primary images seem uninterpretable and unconvincing in supporting the authors' claims. There is a lack of clear evidence in support of TF coexpression and that their expression precedes Tal1.

Figure 4 has been entirely redrawn with higher resolution images and a more logical layout. In the revised Figure 4, only the most relevant ISH images are shown and arrowheads are added showing the colocalization of the mRNA in the cell cytoplasm. Next to the plots of RNA expression along the apical-basal axis of r1, an explanatory image of the quantification process is added (Figure 4D).

(5) What was gained from also performing ChromVAR other than finding more potential regulators and do the results of the two kinds of analyses corroborate one another? What is a dual GATA:TAL BS?

Our motivation for ChromVAR analysis is now more clearly stated in the text (Results, page 9): “In addition to the regulatory elements of GABAergic fate selectors, we wanted to understand the genome-wide TF activity during rV2 neuron differentiation. To this aim we applied ChromVAR (Schep et al., 2017)" Also, further explanation about the Tal1and Gata binding sites has been added in this chapter (Results, page 9).

The dual GATA:Tal BS (TAL1.H12CORE.0.P.B) is a 19-bp motif that consists of an E-box and GATA sequence, and is likely bound by heteromeric Gata2-Tal1 TF complex, but may also be bound by Gata2, Gata3 or Tal1 TFs separately. The other TFBSs of Tal1 contain a strong E-box motif and showed either a lower activity (TAL1.H12CORE.1.P.B) or an earlier peak of activity in common precursors with a decline after differentiation (TAL1.H12CORE.2.P.B) (Results, page 9).

(6) The way the data are displayed it is difficult to see how the C&T confirmed the binding of Ebf1 and Insm1, Tal1, Gata2, and Gata3 (Supplementary Figures 9-11). Are there strong footprints (scores) centered at these peaks? One can't assess this with the way the displays are organized in Figure 3. What is the importance of the H3K4me3 C&T? Replicate consistency, while very strong for some TFs, seems low for other TFs, e.g. Vsx2 C&T on Tal1 and Gata2. The overlaps do not appear very strong in Supplementary Figure 10. Panels are not letter labeled.

We have added an analysis of footprint locations within the CUT&Tag peaks (Supplementary Figure 11). The Figure shows that the footprints are enriched at the middle regions of the CUT&Tag peaks, which is expected if TF binding at the footprinted TFBS site was causative for the CUT&Tag peaks.

The aim of the Supplementary Figures 9-11 (Supplementary Figures 8-10 in the revised manuscript) was to show the quality and replicability of the CUT&Tag.

The anti-H3K4me3 antibody, as well as the anti-IgG antibody, was used in CUT&Tag as part of experiment technical controls. A strong CUT&Tag signal was detected in all our CUT&Tag experiments with H3K4me3. The H3K4me3 signal was not used in downstream analyses.

We have now labelled the H3K4me3 data more clearly as "positive controls" in the Supplementary Figure 8. The control samples are shown only on Supplementary Figure 8 and not in the revised Supplementary Figure 10, to avoid repetition. The corresponding figure legends have been modified accordingly.

To show replicate consistency, the genome view showing the Vsx2 CUT&Tag signal at *Gata2* gene has been replaced by a more representative region (Supplementary Figure 8, Vsx2). The Vsx2 CUT&Tag signal at the *Gata2* locus is weak, explaining why the replicability may have seemed low based on that example.

Panel labelling is added on Supplementary Figures S8, S9, S10.

(7) It would be illuminating to present 1-2 detailed examples of specific target genes fulfilling the multiple criteria outlined in Methods and Figure 6A.

We now present examples of the supporting evidence used in the definition of selector gene target features and target genes. The new Supplementary Figure 12 shows an example gene *Lmo1* that was identified as a target gene of Tal1, Gata2 and Gata3.

**Reviewer #2 (Recommendations for the authors):**
(1) The authors perform CUT&Tag to ask whether Tal1 and other TFs indeed bind putative CREs computed. However, it is unclear whether some of the antibodies (such as Gata3, Vsx2, Insm1, Tead2, Ebf1) used are knock-out validated for CUT&Tag or a similar type of assay such as ChIP-seq and therefore whether the peaks called are specific. The authors should either provide specificity data for these or a reference that has these data. The Vsx2 signal in Figure S9 looks particularly unconvincing.

Information about the target specificity of the antibodies can be found in previous studies or in the product information. The references to the studies have been now added in the Methods (Materials and Methods, CUT&Tag, pages 18-19). Some of the antibodies are indeed not yet validated for ChIP-seq, Cut-and-run or CUT&Tag. This is now clearly stated in the Materials and Methods (page 19): "The anti-Ebf1, anti-Tal1, anti-IgG and anti-H3K4me3 antibodies were tested on Cut-and-Run or ChIP-seq previously (Boller et al., 2016b; Courtial et al., 2012) and Cell Signalling product information. The anti-Gata2 and anti-Gata3 antibodies are ChIP-validated [(Ahluwalia et al., 2020a) and Abcam product information]. There are no previous results on ChIP, ChIP-seq or CUT&Tag with the anti-Insm1, anti-Tead2 and anti-Vsx2 antibodies used here. The specificity and nuclear localization have been demonstrated in immunohistochemistry with anti-Vsx2 (Ahluwalia et al., 2020b) and anti-Tead2 (Biorbyt product information). We observed good correlation between replicates with anti-Insm1, similar to all antibodies used here, but its specificity to target was not specifically tested". We admit that specificity testing with knockout samples would increase confidence in our data. However, we have observed robust signals and good replicability in the CUT&Tag for the antibodies shown here.

Vsx2 CUT&Tag signal at the loci previously shown in Supplementary Figure S9 (now Supplementary Figure 8) is weak, explaining why the replicability may seem low based on those examples. The genome view showing the Vsx2 CUT&Tag signal at Gata2 gene locus in Supplementary Figure 8 (previously Supplementary figure 9) has now been replaced by a view of Vsx2 locus that is more representative of the signal.

(2) It is unclear why the authors chose to focus on the transcription factor genes described in line 626 as opposed to the many other putative TFs described in Figure 3/Supplementary Figure 8. This is the major challenge of the paper - the authors are trying to tell a very targeted story but they show a lot of different names of TFs and it is hard to follow which are most important.

We agree with the reviewer that the process of selection of the genes of interest is not always transparent. We are aware that interpretations of a paper are based on the known functions of the putative regulatory TFs, however additional aspects of regulation could be revealed even if the biological functions of all the TFs were known. This is now stated in the Discussion “Caveats of the study” chapter. It would be relevant to study all identified candidate genes, but as often is the case, our possibilities were limited by the availability of materials (probes, antibodies), time, and financial resources. In the revised manuscript, we now briefly describe the biological processes related to the selected candidate regulatory TFs of the *Tal1* gene (Results, page 8, "Pattern of expression of the putative regulators of *Tal1* in the r1"). We hope this justifies the focus on them in our RNA co-expression analysis. The TFs analysed by RNAscope ISH are examples, which demonstrate alignment of the tissue expression patterns with the scRNA-seq data, suggesting that the dynamics of gene expression detected by scRNA-seq generally reflects the pattern of expression in the developing brainstem.

(3) How is the RNA expression level in Figure 5B and 4D-L computed? These are the clusters defined by scATAC-seq. Is this an inferred RNA expression? This should be made more clear in the text.

The charts in Figures 5B and 4G,H,I show inferred RNA expression. The Y-axis labels have now been corrected and include the term inferred’. RNA expression in the scATAC-seq cell clusters is inferred from the scRNA-seq cells after the integration of the datasets.

(4) The convergence of the GABA TFs on a common set of target genes reminds me of a nice study from the Rubenstein lab PMID: 34921112 that looked at a set of TFs in cortical progenitors. This might be a good comparison study for the authors to use as a model to discuss the convergence data.

We thank the reviewer for bringing this article to our attention. The article is now discussed in the manuscript (Discussion, page 11).

(5) The data in Figure 4, the in-situ figure, needs significant work. First, the images especially B, F, and J appear to be of quite low resolution, so they are hard to see. It is unclear exactly what is being graphed in C, G, and K and it does not seem to match the text of the results section. Perhaps better labeling of the figure and a more thorough description will make it clear. It is not clear how D, H, and L were supposed to relate to the images - presumably, this is a case where cell type is spatially organized, but this was unclear in the text if this is known and it needs to be more clearly described. Overall, as currently presented this figure does not support the descriptions and conclusions in the text.

Figure 4 has been entirely redrawn with higher resolution images and more logical layout. In the revised Figure 4, the ISH data and the quantification plots are better presented; arrows showing the colocalization of the mRNA in the cell cytoplasm were added; and an explanatory image of the quantification process is added on (D).

Minor points(1) Helpful if the authors include scATAC-seq coverage plots for neuronal subtype markers in Figure 1/S1.

We are unfortunately uncertain what is meant with this request. Subtype markers in Figure 1/S1 scATAC-seq based clusters are shown from inferred RNA expression, and therefore these marker expression plots do not have any coverage information available.

(2) The authors in line 429 mention the testing of features within TADs. They should make it clear in the main text (although tadmap is mentioned in the methods) that this is a prediction made by aggregating HiC datasets.

Good point and that this detail has been added to both page 3 and 16.

(3) The authors should include a table with the phastcons output described between lines 511 and 521 in the main or supplementary figures.

We have now clarified int the text that we did not recalculate any phastcons results, we merely used already published and available conservation score per nucleotide as provided by the original authors (Siepel et al. 2005). (Results, page 5: revised text is " To that aim, we used nucleotide conservation scores from UCSC (Siepel et al., 2005). We overlaid conservation information and scATAC-seq features to both validate feature definition as well as to provide corroborating evidence to recognize cCRE elements.")

(4) It is very difficult to read the names of the transcription factor genes described in Figure 3B-D and Supplementary Figure 8 - it would be helpful to resize the text.

The Figures 3B-D and Supplementary Figure 7 (former Supplementary figure 8) have been modified, removing unnecessary elements and increasing the size of text.

(5) It is unclear what strain of mouse is used in the study - this should be mentioned in the methods.

Outbred NMRI mouse strain was used in this study. Information about the mouse strain is added in Materials and Methods: scRNA-seq samples (page 14), scATAC-seq samples (page 15), RNAscope in situ hybridization (page 17) and CUT&Tag (page 18).

(6) Text size in Figure 6 should be larger. R-T could be moved to a Supplementary Figure.

The Figure 6 has been revised, making the charts clearer and the labels of charts larger. The Figure 6R-S have been replaced by Supplementary table 8 and the Figure 6T is now shown as a new Figure (Figure 7).

Additional corrections in figures

Figure 6 D,I,N had wrong y-axis scale. It has been corrected, though it does not have an effect on the interpretation of the data as Pos.link and Neg.link counts were compared to each other’s (ratio).

On Figure 2B, the heatmap labels were shifted making it difficult to identify the feature name per row. This is now corrected.